# GmBTB/POZ promotes the ubiquitination and degradation of LHP1 to regulate the response of soybean to *Phytophthora sojae*

Chuanzhong Zhang[1,2,4], Qun Cheng[1,4], Huiyu Wang[1,4], Hong Gao[1], Xin Fang[1], Xi Chen[1], Ming Zhao[1], Wanling Wei[1], Bo Song[1], Shanshan Liu[1], Junjiang Wu[3], Shuzhen Zhang[1✉] & Pengfei Xu[1✉]

*Phytophthora sojae* is a pathogen that causes stem and root rot in soybean (*Glycine max* [L.] Merr.). We previously demonstrated that GmBTB/POZ, a BTB/POZ domain-containing nuclear protein, enhances resistance to *P. sojae* in soybean, via a process that depends on salicylic acid (SA). Here, we demonstrate that GmBTB/POZ associates directly with soybean LIKE HETEROCHROMATIN PROTEIN1 (GmLHP1) in vitro and in vivo and promotes its ubiquitination and degradation. Both overexpression and RNA interference analysis of transgenic lines demonstrate that GmLHP1 negatively regulates the response of soybean to *P. sojae* by reducing SA levels and repressing *GmPR1* expression. The WRKY transcription factor gene, *GmWRKY40*, a SA-induced gene in the SA signaling pathway, is targeted by GmLHP1, which represses its expression via at least two mechanisms (directly binding to its promoter and impairing SA accumulation). Furthermore, the nuclear localization of GmLHP1 is required for the GmLHP1-mediated negative regulation of immunity, SA levels and the suppression of *GmWRKY40* expression. Finally, GmBTB/POZ releases GmLHP1-regulated *GmWRKY40* suppression and increases resistance to *P. sojae* in *GmLHP1-OE* hairy roots. These findings uncover a regulatory mechanism by which GmBTB/POZ-GmLHP1 modulates resistance to *P. sojae* in soybean, likely by regulating the expression of downstream target gene *GmWRKY40*.

[1] Soybean Research Institute, Northeast Agricultural University, Key Laboratory of Soybean Biology of Chinese Education Ministry, Harbin, China. [2] Key Laboratory of Soybean Molecular Design Breeding, Northeast Institute of Geography and Agroecology, Chinese Academy of Sciences, Harbin, China. [3] Soybean Research Institute of Heilongjiang Academy of Agricultural Sciences, Key Laboratory of Soybean Cultivation of Ministry of Agriculture, Harbin, China. [4] These authors contributed equally: Chuanzhong Zhang, Qun Cheng, Huiyu Wang. ✉email: zhangshuzhen@neau.edu.cn; xupengfei@neau.edu.cn

Plants have sophisticated cell-autonomous defense mechanisms that combat microbial pathogens, including a waxy cuticle, anti-microbial compounds, and plant innate immunity systems[1,2]. In general, the waxy cuticle and preformed anti-microbial compounds provide passive protection against pathogens rather than attacking a specific host[1], whereas plants rely on innate immunity to defend themselves against widespread diseases[3,4]. These immunity responses arise via a regulatory network coordinating immune response proteins, transcriptional regulators, and other structural components[5–7]. Regulation occurs at every level, from differential transcript accumulation and processing to protein modification and turnover[5,8]. Thus, research on the regulatory components of plant defense responses can provide insights into the complex processes involved in plant immunity.

Ubiquitination is a common post-translational modification in which ubiquitin (Ub) is covalently bound to lysine residues in target proteins[9,10]. Ubiquitination is carried out by Ub-activating (E1), Ub-conjugating (E2), and Ub-ligase (E3) enzymes, and often leads to target protein degradation mediated by the 26S proteasome[11]. The BTB/POZ domain (Broad Complex, Tramtrack, Bric-a-brac/Pox virus and Zinc finger) is an evolutionarily conserved, $NH_3$-terminal protein–protein interaction motif present in a variety of cytoskeletal modifiers and Ub ligase substrate recognition factors[12–14]. Substrate specificity factors associate with cullin 3-based E3 ligases through BTB/POZ proteins[15]. Therefore, BTB/POZ proteins function as a bridge between CRL3 (CUL3-RING E3 ligase) and substrate proteins and are essential for the ubiquitination process[16,17].

HP1 (HETEROCHROMATIN PROTEIN1) was first described in Drosophila melanogaster as a non-histone chromosomal protein that preferentially binds to constitutive heterochromatin on polytene chromosomes[18]. HP1 orthologs are present in organisms ranging from yeasts to humans[19,20]. Plants possess a single-copy gene for HP1, LIKE HETEROCHROMATIN PROTEIN1 (LHP1)[21], which was initially identified in screens for inflorescence meristem function in Arabidopsis thaliana and is also referred to as TERMINAL FLOWER2 (refs. [22,23]). To date, many plant LHP1 homologs have been identified[24–26]. LHP1 encodes a highly evolutionarily conserved protein containing a chromo domain and a chromo shadow domain[21,24].

LHP1 proteins regulate several important growth and development processes in plants[27,28]. Mutations in AtLHP1 cause a range of developmental defects, including reduced stability of the vernalized state, conversion of the shoot apical meristem to a terminal flower, curled leaves, and reduced root growth[21,29]. LHP1 is also involved in auxin biosynthesis in Arabidopsis[30]. In general, LHP1 proteins also function as transcriptional repressors, which play crucial roles in maintaining the transcriptionally silenced state of their targets[31–33]. For example, AtLHP1 directly represses the expression of the floral promoter FLOWERING LOCUS T (FT) in vascular tissue before dusk and at night[34]. The early-flowering phenotype of Arabidopsis lhp1 mutants results from increased expression of FT[23]. These findings indicate that LHP1 represses the transcription of genes that function during different stages of reproductive development. Nevertheless, most studies of LHP1 performed to date in plants other than Arabidopsis were limited to examining the differences in protein expression profiles, whereas no in-depth study of gene expression, functions, or molecular mechanisms of plant LHP1s have been performed. In particular, the role of LHP1 in soybean (Glycine max [L.] Merr.) in response to biotic stress has not yet been evaluated.

GmBTB/POZ positively regulates the response of soybean to Phytophthora sojae, a destructive pathogen that causes stem and root rot in soybean; this response primarily depends on the salicylic acid (SA) signaling pathway[35]. In the current study, we focused on soybean LIKE HETEROCHROMATIN PROTEIN1 (GmLHP1; NCBI protein no. XP_003548606), a GmBTB/POZ-interacting partner involved in the response to P. sojae infection. GmLHP1 was degraded in soybean inoculated with P. sojae, primarily through the 26S proteasome. Further analysis showed that GmBTB/POZ promotes the ubiquitination and degradation of GmLHP1 in vitro and in vivo. In addition, GmLHP1 inhibits the expression of GmWRKY40, a SA-inducible gene that functions downstream of SA biosynthesis. Therefore, we uncovered a potential role of the GmBTB/POZ–GmLHP1 regulatory module in plant pathogen resistance, providing insights into the mechanism underlying defense responses against P. sojae infection in soybean.

## Results

**GmLHP1 interacts with GmBTB/POZ.** We previously demonstrated that GmBTB/POZ positively regulates the response of soybean to P. sojae infection and GmBTB/POZ interacted with GmLHP1 (LIKE HETEROCHROMATIN PROTEIN1) in a bimolecular fluorescence complementation (BiFC) assay[35]. In soybean, there are two genes encoding copies of LHP1 (LHP1-1 and LHP1-2)[36]. In the current study, we focused on LHP1-1, namely GmLHP1 (NCBI protein no. XP_003548606; Glyma.16G079900) which contains two highly conserved structural domains: a chromo domain and a chromo shadow domain (Supplementary Fig. 1). Firstly, in a Y2H assay, yeast cells co-expressing pGBD-GmLHP1 + pGAD-GmBTB/POZ or pGBD-GmBTB/POZ + pGAD-GmLHP1, but not pGBD-GmLHP1 + pGAD or pGBD-GmBTB/POZ + pGAD, grew well on SD/-Trp/-Leu/-His/-Ade (QDO) screening medium and showed α-galactosidase activity (Fig. 1a), indicating that GmLHP1 interacts with GmBTB/POZ in yeast cells.

We performed an in vitro pull-down assay to validate the interaction between GmLHP1 and GmBTB/POZ. GmLHP1-His, GmBTB/POZ-GST, and GST alone were detected in whole-cell lysates (Input). GmLHP1 fused with a His tag was not detected in the control sample (GST protein alone), whereas GmLHP1-His was pulled down via GmBTB/POZ-GST (Fig. 1b), suggesting that GmLHP1 directly interacts with GmBTB/POZ. We further confirmed the interaction between GmLHP1 and GmBTB/POZ using firefly luciferase complementation imaging (LCI). The results confirmed that GmLHP1 interacts with GmBTB/POZ in planta (Fig. 1c). Furthermore, these assays indicated that GmLHP1 interacts with GmBTB/POZ in the nucleus (Fig. 1c). Therefore, these three different methods indicated that GmLHP1 directly interacts with GmBTB/POZ both in vitro and in vivo.

**GmBTB/POZ promotes the ubiquitination and degradation of GmLHP1.** BTB/POZ proteins are a bridge between CUL3-RING E3 ligase and substrate proteins, and they are essential for the Ub process[17,37]. Since our protein interaction assays between GmLHP1 and GmBTB/POZ suggested that GmLHP1 is a potential substrate of GmBTB/POZ, we speculated that GmBTB/POZ plays a role in the ubiquitination and degradation of GmLHP1. To explore this possibility, we performed in vitro protein degradation assays. Specifically, protein extracts from the WT soybean were incubated with the His-tagged GmLHP1 (GmLHP1-His) proteins purified from Escherichia coli Rosetta (DE3) cells at 22 °C. Then, we performed an immunoblot assay using anti-His antibody to measure the abundance of GmLHP1-His protein. GmLHP1-His was unstable in WT soybean protein extracts; clear GmLHP1-His degradation was observed beginning at 0.5 h, and it was almost completely degraded by 3 h (Fig. 2a). However, treating the samples with 100 μM of the proteasome inhibitor MG132 significantly repressed the degradation process (Fig. 2a). This observation suggests that GmLHP1 is normally

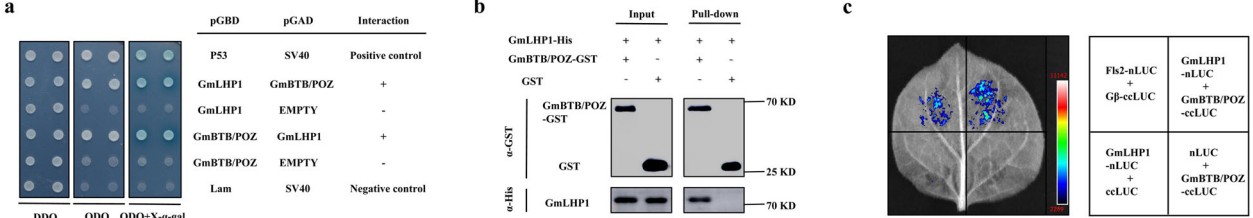

**Fig. 1 GmLHP1 interacts with GmBTB/POZ. a** GmLHP1 interacts with GmBTB/POZ in yeast cells. The yeast cells were selected on SD medium lacking Leu and Trp (DDO), and interaction was assessed based on their ability to grow on selective SD medium lacking Leu, Trp, His, and Ade (QDO) or SD medium lacking Leu, Trp, His, and Ade (QDO) but containing X-α-Gal for 3 days at 30 °C. The combination of pGBD-P53 + pGAD-SV40 was used as a positive control and pGBD-Lam + pGAD-SV40 as a negative control. X-α-Gal represents 5-bromo-4-chloro-3-indolyl-α-D-galactoside. **b** In vitro pull-down assays showing the interactions of GmLHP1 with GmBTB/POZ. His-tagged proteins were incubated with immobilized GST or GST-tagged proteins, and immunoprecipitated fractions were detected by anti-His antibody. **c** Interaction between GmLHP1 and GmBTB/POZ in LCI assays. The combination of Fls2-nLUC + Gβ-ccLUC was used as a positive control.

degraded, and it points to the possible involvement of the 26S proteasome pathway and ubiquitination.

To determine whether GmBTB/POZ improves the ubiquitination and degradation of GmLHP1, we identified three T4 *GmBTB/POZ*-OE soybean plants and three T4 *GmBTB/POZ*-RNAi soybean plants using immunoblot analysis, QuickStix Kit for LibertyLink (bar) strips, and quantitative reverse-transcription PCR (qRT-PCR), respectively (Supplementary Fig. 2a–d) and subjected them to degradation experiments. Compared to the WT, the degradation rate of GmLHP1-His significantly increased in *GmBTB/POZ*-OE plant extract (Fig. 2c–e), whereas its stability increased in *GmBTB/POZ*-RNAi extract (Fig. 2g–i). These results indicate that GmBTB/POZ enhances the degradation of GmLHP1 in vitro. To further explore this notion, we performed in vivo ubiquitination assays. We transformed soybean hairy roots with the plant binary expression vector system *p35S: Flag-GmLHP1* + *p35S: GmBTB/POZ-Myc* (Fig. 2j), immunoprecipitated GmLHP1-Flag and GmBTB/POZ-Myc from proteins extracted from the plants using anti-Flag antibody, and probed the eluted proteins with anti-Flag and anti-Ubi antibodies. In hairy roots overexpressing *GmBTB/POZ*, much more ubiquitinated GmLHP1-Flag protein was detected compared to the WT (Fig. 2k). Taken together, these results indicate that GmBTB/POZ likely promotes the ubiquitination of GmLHP1 in vitro and in vivo.

To explore the active region of GmBTB/POZ involved in interaction of GmLHP1, different regions of the *GmBTB/POZ* cDNA encoding the full length (amino acids 1–258), the N-terminal part (amino acids 1–83), the BTB/POZ domain part (amino acids 84–188), or the C-terminal part (amino acids 189–258) of the protein were inserted into yeast vectors pGBD (Supplementary Fig. 3a, b). The Y2H assay showed that both the full-length, the N-terminal part (amino acids 1–83) and the C-terminal part (amino acids 189–258) of the GmBTB/POZ protein were able to interact with the GmLHP1 protein, but not the BTB/POZ domain part (amino acids 84–188) of GmBTB/POZ (Supplementary Fig. 3b). Consistent results were obtained by the BiFC assay (Supplementary Fig. 3c). Furthermore, it is reported that members of the BTB/POZ protein family use the BTB/POZ domain to bind the cullin-based E3 ligases and the other regions to recruit substrate proteins[38–40]. In this study, the interaction analyses suggested that the N-terminal or the C-terminal of the GmBTB/POZ may play a role in recruiting GmLHP1 in GmBTB/POZ-mediated ubiquitination. To test the hypothesis, we firstly constructed the (domain+C) and (N+domain) regions of GmBTB/POZ and verified that both the (domain+C) and (N+domain) regions still interact with GmLHP1 by the Y2H assay and BiFC (Supplementary Fig. 3e–g). Then, we further analyzed that whether the (domain+C) and (N+domain) region proteins can still function as a bridge in the ubiquitination of GmLHP1 by

in vitro cell-free degradation assay and in vivo ubiquitination assay. The results showed that both the (domain+C) and (N+domain) region proteins can promote the ubiquitination and degradation of GmLHP1 in vitro and in vivo (Supplementary Fig. 3d, h). These findings suggest that the N-terminal and C-terminal of GmBTB/POZ could be the active region of GmBTB/POZ involved in interaction and recruitment of GmLHP1 in the protein ubiquitination.

**GmLHP1 negatively regulates plant immunity**. LHP1 plays an important role in plant responses to environmental stimuli[41]. In addition, LHP1 interacts with various proteins to perform distinct roles in different cell types[42,43]; for example, LHP1 interacts with GmPHD6 to regulate the expression of genes involved in salt tolerance[36]. GmBTB/POZ plays an integral role in the response of soybean to *P. sojae* attack, which mainly depends on the SA signaling pathway[35]. The interaction and ubiquitination between GmBTB/POZ and GmLHP1 raised the question of whether GmLHP1 plays a role in a GmBTB/POZ-mediated SA and immune signaling pathway. To explore the biological function of GmLHP1, we produced transgenic soybean plants expressing *p35S: Flag-GmLHP1* (*GmLHP1OE*) or *p35S: GmLHP1*-RNA interference (*GmLHP1RNAi*). Immunoblotting analysis confirmed the expression of recombinant GmLHP1-Flag protein in three independently selected T4 *GmLHP1*-OE lines (*GmLHP1OE5*, *GmLHP1OE10*, and *GmLHP1OE14*) using anti-Flag antibody (Supplementary Fig. 2e). Southern blot and qRT-PCR analyses confirmed the reduced expression of *GmLHP1* in the three independently selected T4 *GmLHP1*-RNAi transgenic soybean lines (*GmLHP1RNAi4*, *GmLHP1RNAi5*, and *GmLHP1RNAi6*), which were integrated into the genomes of the three lines in a single copy (Supplementary Fig. 2g, h).

We investigated *P. sojae* resistance in the roots of these transgenic plants. At 96 h of post-inoculation (hpi), the roots of all three *GmLHP1OE* soybean lines exhibited more serious symptoms than WT roots, including watery and rotting lesions (Fig. 3a). By contrast, the three *GmLHP1RNAi* soybean lines displayed almost no visible lesions compared to WT roots (Fig. 3a). We analyzed the relative biomass of *P. sojae* in soybean roots based on the transcript level of *P. sojae* TEF1 (EU079791). *P. sojae* biomass was significantly higher (**$P < 0.01$) in the *GmLHP1OE* lines and significantly lower (**$P < 0.01$) in the *GmLHP1RNAi* lines compared to WT plants (Fig. 3b). Similar results were obtained for *GmLHP1-OE* and *GmLHP1-RNAi* transgenic soybean hairy roots, which were generated by high-efficiency *Agrobacterium rhizogenes*-mediated transformation (Fig. 3f, g, j)[44,45]. These results indicate that overexpressing *GmLHP1* in soybean increases susceptibility to *P. sojae* and that silencing this gene improves resistance to *P. sojae*.

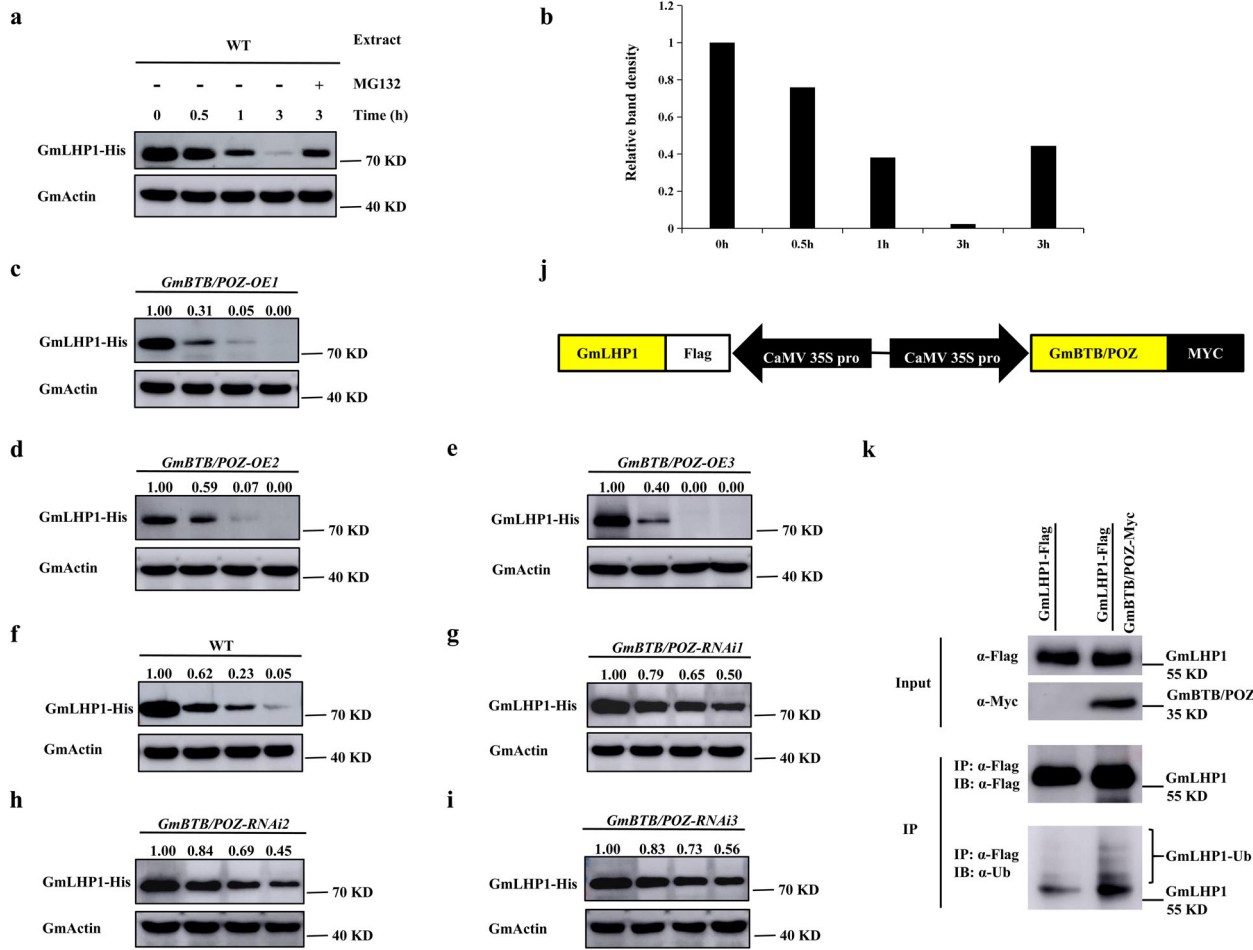

**Fig. 2 GmBTB/POZ promotes the ubiquitination and degradation of GmLHP1. a** GmLHP1 protein was degraded, which likely occurred primarily through the 26S proteasome. For MG132 treatment, WT soybean plant extracts were treated with 100 μM MG132 for 1 h and incubated with GmLHP1-His protein for the indicated time. GmActin was used as a loading control. **b** Relative band density of GmLHP1-His. GmLHP1-His was quantified using ImageJ software. **c–i** In vitro cell-free degradation assays of GmLHP1-His in protein extracts from *GmBTB/POZ* transgenic soybean plants. Protein extracts from transgenic (*GmBTB/POZ-OE* and *GmBTB/POZ-RNAi*) and WT soybean plants were incubated with GmLHP1-His for the indicated time. GmLHP1-His levels were visualized by immunoblotting using anti-His antibody. GmActin was used as a loading control. The protein level of 0 h was set to 1.00. **j** Diagram of the plant binary expression vector system (*p35S: Flag-GmLHP1 + p35S: GmBTB/POZ-Myc*). **k** GmBTB/POZ promotes the ubiquitination of GmLHP1 in vivo. GmLHP1-Flag was immunoprecipitated using anti-Flag-Tag Mouse mAb (Agarose Conjugated) from *GmLHP1-OE* and *GmLHP1-OE/GmBTB/POZ-OE* transgenic soybean hairy roots by high-efficiency *A. rhizogenes*-mediated transformation. The transgenic hairy roots were treated with 100 μM MG132 for 8 h before extraction. The immunoprecipitated protein was examined using anti-Flag and anti-ubi antibodies.

SA plays major roles in regulating basal defense responses during plant immunity[46] and acts as a crucial signaling element in systemic acquired resistance (SAR) signaling pathways[47,48]. SA mediates SAR, which limits the growth of biotrophic and necrotrophic virulent pathogens and favors long-term protection against a broad spectrum of microorganisms[49,50]. Increased endogenous SA levels trigger SAR by inducing the expression of pathogenesis-related (PR) genes, such as *PR1*, which is considered to be an effector gene for SAR[48]. To determine whether GmLHP1 also regulates the SA signaling pathway, we analyzed the SA contents and expression levels of *GmPR1* (AF136636) in *GmLHP1OE*, WT, and *GmLHP1RNAi* soybean plants. Both SA levels and *GmPR1* expression levels were significantly lower (**$P < 0.01$) in *GmLHP1OE* plants and higher (**$P < 0.01$) in *GmLHP1RNAi* plants compared to WT (Fig. 3c, d). In addition, both SA levels and *GmPR1* expression levels were significantly reduced (**$P < 0.01$) in *GmLHP1-OE* transgenic hairy roots vs. the control. However, SA levels and *GmPR1* expression levels were significantly higher (**$P < 0.01$) in *GmLHP1-RNAi* vs. control hairy roots (Fig. 3h, i). The results suggest that GmLHP1

regulates defense responses against *P. sojae* by affecting SA levels and *GmPR1* expression.

**GmLHP1 regulates the transcription of *GmWRKY40* via two mechanisms.** LHP1 is a nucleus-localized protein that generally functions as a transcriptional repressor in both plants and animals[21,33,34,51]. To examine the subcellular localization of GmLHP1, we analyzed the expression of the GmLHP1-GFP fusion protein. GmLHP1-GFP signals were observed in the nuclei of transformed cells, like the GmBTB/POZ-GFP expression pattern reported by Zhang et al.[35], indicating that GmLHP1 is a nucleus-localized protein (Supplementary Fig. 4a). In a transient expression assay in yeast cells using a GAL4-responsive reporter system, transformed yeast cells containing DBD-GmLHP1 (pGBKT7-GmLHP1) exhibited no α-gal activity, indicating that GmLHP1 did not activate the transcription of the reporter gene (Supplementary Fig. 4b).

To explore the extent of the regulatory impact of GmLHP1 and to identify GmLHP1-regulated genes, we performed RNA-sequence (RNA-Seq) analysis of the transcriptomes of both WT

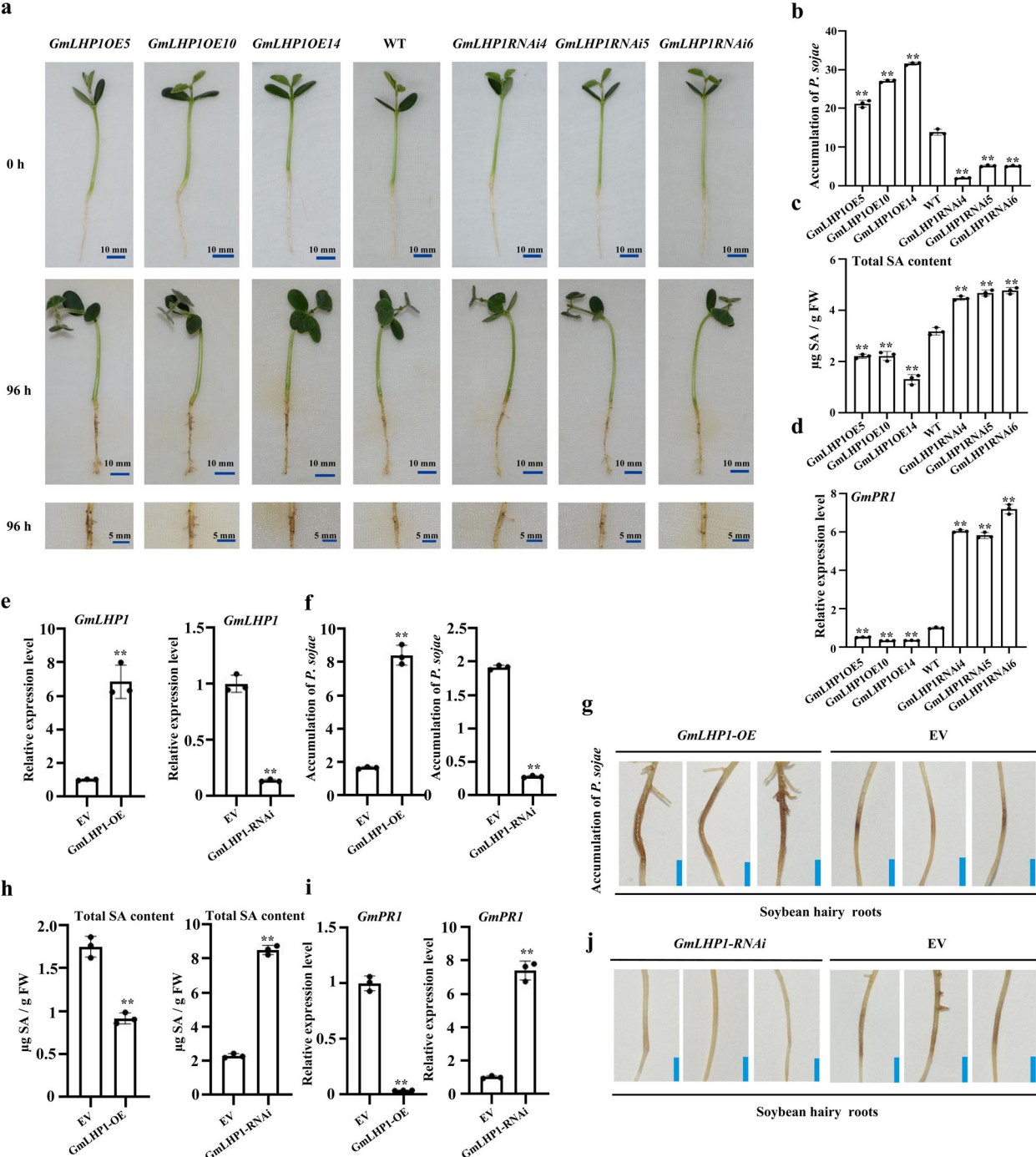

**Fig. 3 GmLHP1 negatively regulates plant immunity. a** Disease symptoms in the roots of wild-type (WT), *GmLHP1OE*, and *GmLHP1RNAi* soybean plants at 96 h after inoculation with *P. sojae*. **b** Relative biomass of *P. sojae* in WT, *GmLHP1OE*, and *GmLHP1RNAi* soybean plants based on *P. sojae TEF1* (EU079791) transcript levels. **c** SA contents in leaves of transgenic and WT soybean. FW, fresh weight. **d** Relative *GmPR1* expression levels in transgenic and WT soybean plants. The expression level of the control sample (WT plants) was set to 1. **e** qRT-PCR analysis of relative *GmLHP1* expression in transgenic soybean hairy roots. Empty vector (EV) transgenic hairy roots were used as controls, and the expression level of the control sample (EV) was set to 1. **f** Relative biomass of *P. sojae* in *GmLHP1*-transgenic hairy roots based on *P. sojae TEF1* (EU079791) transcript levels. **g** Typical infection symptoms of *GmLHP1-OE* and EV soybean hairy roots at 48 h after *P. sojae* inoculation. Bars, 0.5 cm. **h** SA contents in *GmLHP1-OE*, *GmLHP1-RNAi*, and EV hairy roots. FW, fresh weight. **i** Relative *GmPR1* expression levels in *GmLHP1-OE*, *GmLHP1-RNAi*, and EV hairy roots. The expression level of the control sample (EV) was set to 1. **j** Typical infection symptoms of *GmLHP1-RNAi* and EV hairy roots at 48 h after *P. sojae* inoculation. Bars, 0.5 cm. The housekeeping gene *GmEF1* was used as an internal control to normalize the data. The experiment was performed on three biological replicates, each with three technical replicates, and the results were statistically analyzed using Student's *t*-test (*$P < 0.05$, **$P < 0.01$). Bars indicate the standard deviation of the mean ($n = 3$).

and *GmLHP1OE* transgenic soybean plants after 6 weeks of growth in the field. RNA-Seq analysis identified 422 differentially expressed genes (DEGs) with >2.0-fold differences in expression in *GmLHP1*-OE vs. WT plants under non-stress conditions (false discovery rate (FDR) **$P < 0.01$). Among the 422 DEGs, 253 were significantly upregulated and 169 were significantly down-regulated (Supplementary Fig. 5a and Fig. 4a). Gene ontology (GO) analysis revealed that these genes are primarily enriched in the GO terms plant response to biotic and abiotic stress, hormone stimulus, transferase activity, transport, and other metabolic processes (Supplementary Fig. 5b).

We examined the expression of several downregulated stress-related DEGs in *GmLHP1OE* and *GmLHP1RNAi* soybean plants by qRT-PCR analysis. Examples of these genes include immunity signaling genes such as *GmMEKK2* (Glyma.17G173000), *GmWRKY40* (Glyma.15G003300), and *GmCPK2* (Glyma.11G206300) and defense-associated genes such as *GmNAC90* (Glyma.11G182000), *GmNAC29* (Glyma.02G109800), *GmERF104* (Glyma.20G070000), *GmbHLH35* (Glyma.13G101100), *GmMYB70* (Glyma.17G237900), and *GmMLP34* (Glyma.09G102400). *GmWRKY40* expression was dramatically reduced in *GmLHP1OE* vs. WT plants. Notably, in *GmLHP1RNAi* soybean plants, *GmWRKY40* expression significantly increased (**$P < 0.01$) compared to the WT, while none of the other genes showed markedly altered expression (Fig. 4b). These findings indicate that the regulation of *GmWRKY40* expression likely plays a role in GmLHP1-mediated defense responses.

To explore how GmLHP1 regulates the expression of *GmWRKY40*, we performed a dual effector–reporter assay using GmLHP1 as the effector and the luciferase gene under the control of 2.0 kb of the *GmWRKY40* promoter as the reporter. The effector construct harbored *GmLHP1* expressed under the control of the 35S promoter (*p35S: Flag-GmLHP1*). We transformed the reporter construct (*p35S: REN-pGmWRKY40: LUC*) and the effector construct (*p35S: Flag-GmLHP1*) or the reporter construct (*p35S: REN-pGmWRKY40: LUC*) and the blank effector construct (empty vector (EV)) into healthy *Nicotiana benthamiana* leaves. The co-existence of *GmLHP1* and the *GmWRKY40* promoter significantly inhibited (**$P < 0.01$) luciferase expression in *N. benthamiana* leaves (Fig. 4e, f), suggesting that GmLHP1 significantly represses the expression of *GmWRKY40*.

To investigate the binding capacity of GmLHP1 to the promoter of *GmWRKY40*, we performed chromatin immuno-precipitation (ChIP)-qPCR assays using cell extracts from WT plants and *GmLHP1OE* transgenic soybean plants expressing Flag-fused GmLHP1 under the control of the constitutive 35S promoter. However, since the binding elements or target regions of LHP1 were unclear, we analyzed the enrichment of four regions in the *GmWRKY40* promoter via ChIP-qPCR using four pairs of specific primers. As shown in Fig. 4c, d, the *c* region was significantly enriched (**$P < 0.01$) with GmLHP1-Flag, whereas none of the three regions in the *GmEF1* promoter were enriched with GmLHP1-Flag; *GmEF1* is often used as a reference gene, since it is expressed constitutively at a constant level throughout the plant and is not influenced by exogenous treatment[52]. These results indicate that GmLHP1 specifically associates with the regulatory regions of its target gene (Fig. 4c, d).

Some *WRKY* genes are SA-inducible transcription factor genes involved in disease resistance in a number of plant species[53–55]. Analysis of *GmWRKY40* transcript levels in response to SA (0.5 mM) treatment showed that *GmWRKY40* expression was significantly induced by SA in WT plants, reaching a peak at 12 h, followed by a steep decline (Fig. 4g), indicating that *GmWRKY40* expression is significantly induced by SA in soybean. To further elucidate the underlying regulatory mechanism, we used the 2 kb

promoter region of *GmWRKY40* to drive the expression of the GUS reporter gene in the pBI121 expression vector, which we transformed into "Dongnong 50" soybean hairy roots via high-efficiency *A. rhizogenes*-mediated transformation. We analyzed *GmWRKY40* promoter activity in hairy roots at 6 h after SA treatment. The amount of histochemical GUS staining in hairy roots was higher under SA treatment than under mock ($H_2O$) treatment (Fig. 4h), suggesting that *GmWRKY40* functions downstream of SA biosynthesis. Moreover, we demonstrated that GmLHP1 has an effect on SA accumulation (Fig. 3h).

To investigate whether GmLHP1 suppresses *GmWRKY40* expression via impaired SA accumulation, we examined whether exogenous SA application would weaken the inhibition of *GmWRKY40* expression in *GmLHP1OE* soybean plants. As shown in Fig. 4i, we analyzed the expression efficiency of *GmLHP1* by qRT-PCR. *GmLHP1* transcript levels were significantly higher in *GmLHP1*-OE plants compared to WT plants under both mock ($H_2O$) treatment and after 6 h of SA treatment. As expected, SA-treated plants displayed clearly increased *GmWRKY40* transcript abundance compared to mock-treated plants (Fig. 4j). *GmWRKY40* transcript levels were markedly lower (**$P < 0.01$) in *GmLHP1OE* plants than in WT plants under mock treatment, while there was no obvious difference (*$P < 0.05$) in *GmWRKY40* transcript level between *GmLHP1OE* and WT plants under SA treatment, suggesting that SA induces changes in *GmWRKY40* expression in *GmLHP1OE* soybean plants. Taken together, these findings suggest that at least two mechanisms (direct repression of *GmWRKY40* expression and impaired SA accumulation) contribute to the regulation of *GmWRKY40* expression by GmLHP1.

**GmWRKY40 also functions in responses to *P. sojae* infection and increases the expression of SA-marker gene GmPR1.** We then explored the possible role of *GmWRKY40* in the response to *P. sojae* infection by analyzing the phenotypes of control, *GmWRKY40-OE*, and *GmWRKY40-RNAi* hairy roots after incubation with *P. sojae* zoospores. The *GmWRKY40-OE* transgenic hairy roots were examined by immunoblotting (Supplementary Fig. 2k) and qRT-PCR (Fig. 5c) and the *GmWRKY40-RNAi* transgenic hairy roots by analysis with QuickStix Kit for LibertyLink bar strips (Supplementary Fig. 2l) and qRT-PCR (Fig. 5c). After 2 d of incubation with *P. sojae* zoospores, *GmWRKY40-OE* hairy roots displayed almost no visible lesions (Fig. 5a), whereas the *GmWRKY40-RNAi* lines exhibited enhanced wilting symptoms and chlorosis compared to the control (Fig. 5b). We also analyzed the relative biomass of *P. sojae* in infected hairy roots after 2 days of incubation with *P. sojae* zoospores. The biomass of *P. sojae* was significantly (**$P < 0.01$) lower in the roots of *GmWRKY40-OE* lines but significantly (**$P < 0.01$) higher in the roots of *GmWRKY40-RNAi* lines compared to the control (Fig. 5d).

*GmWRKY40* expression was significantly induced by SA, suggesting that *GmWRKY40* functions downstream of SA biosynthesis as a component of SA signaling. To determine whether *GmWRKY40* also participates in the SA signaling pathway, we measured SA content and *GmPR1* expression in the transgenic hairy roots. *GmPR1* was expressed at significantly higher levels (**$P < 0.01$) in *GmWRKY40-OE* lines but at significantly lower levels (**$P < 0.01$) in *GmWRKY40-RNAi* lines compared to the control (Fig. 5f). However, there was no significant difference in SA level between the *GmWRKY40* lines and control hairy roots (Fig. 5e). These results suggest that *GmWRKY40* functions as a SA-induced gene downstream of SA biosynthesis and enhances the expression of SA-marker gene *GmPR1* in response to *P. sojae* infection.

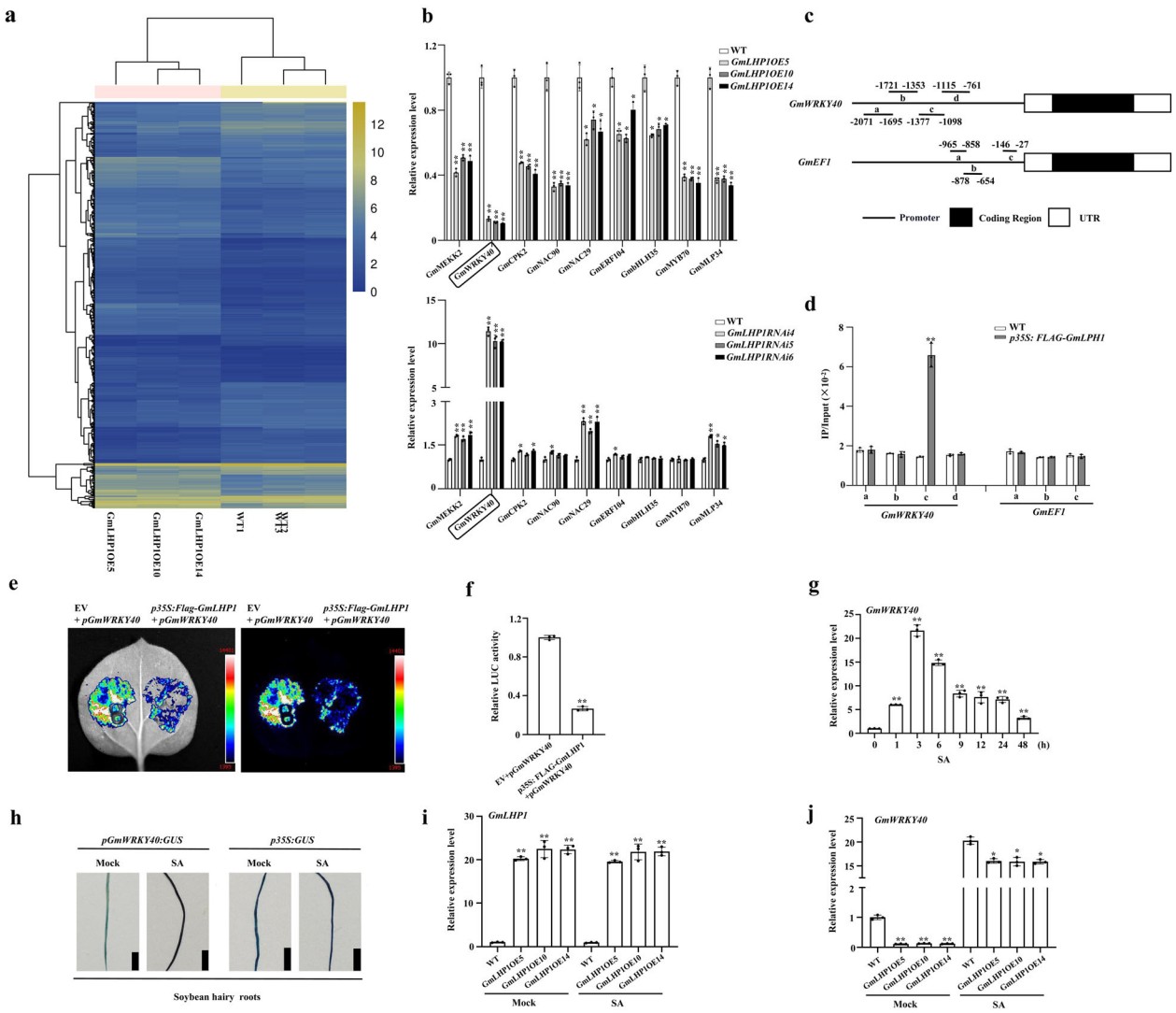

**Fig. 4 *GmWRKY40* is a target gene of GmLHP1. a** Heat map of the expression patterns of significantly differentially expressed genes in WT and *GmLHP1OE* soybean plants determined by RNA-Seq analysis. The scale bar indicates fold changes (log2 value). **b** Relative expression of several stress-related genes in WT, *GmLHP1OE*, and *GmLHP1RNAi* soybean plants. **c**, **d** ChIP analysis of GmLHP1 binding to the *GmWRKY40* promoter region. Chromatin from LHP1-Flag transgenic soybean plants and the WT was immunoprecipitated by anti-Flag or no antibody. The precipitated chromatin fragments were analyzed by qPCR using four pairs of specific ChIP-qPCR primer sets to amplify four regions upstream of *GmWRKY40* (pGmWRKY40a, pGmWRKY40b, pGmWRKY40c, and pGmWRKY40d), as indicated. One-tenth of the input (without antibody precipitation) of chromatin was analyzed and used as a control. *pGmEF1* was used as a negative control. Three biological replicates, each with three technical replicates, were averaged and statistically analyzed using Student's *t*-test (*$P < 0.05$, **$P < 0.01$). Bars indicate standard deviation of the mean ($n = 3$). **e** Dual-luciferase assay in *N. benthamiana* leaves showing that GmLHP1 represses the expression of *GmWRKY40*. Representative photographs are shown. **f** Detection of LUC/REN activity to verify that GmLHP1 represses the transcription of *GmWRKY40*. The combination of the reporter construct (pGmWRKY40: LUC) and the blank effector construct (empty vector) was used as a control. **g** Relative expression of *GmWRKY40* in WT soybean plants in response to SA (0.5 mM) treatment. The relative expression levels of *GmWRKY40* were compared with those of mock-treated plants. Fourteen-day-old plants were used for analysis. **h** *GmWRKY40* promoter-driven GUS expression in transgenic soybean hairy roots under SA or mock treatment for 3 h. Bars, 0.5 cm. **i** Expression patterns of *GmLHP1* in WT and *GmLHP1OE* transgenic soybean plants under SA or mock treatment for 3 h. The expression level of the control sample (mock-treated wild-type (WT) plants) was set to 1. **j** Expression patterns of *GmWRKY40* in WT and *GmLHP1OE* transgenic soybean plants under SA or mock treatment for 3 h. The expression level of the control sample (mock-treated wild-type (WT) plants) was set to 1. The housekeeping gene *GmEF1* was used as an internal control to normalize the data. The experiment was performed on three biological replicates, each with three technical replicates, and the results were statistically analyzed using Student's *t*-test (*$P < 0.05$, **$P < 0.01$). Bars indicate the standard deviation of the mean ($n = 3$).

**Nuclear localization of GmLHP1 is required for its functionality.** To determine the region(s) responsible for the nuclear localization of GmLHP1 and analyze whether the nuclear localization of GmLHP1 is required for its functionality, we firstly analyzed the nuclear localization signal (NLS) regions of GmLHP1 using NLS Mapper software[56,57]. Three putative NLS regions (NLS1 to NLS3) were identified (Fig. 6a, left column), NLS1

(IRRKR-EVQY, amino acids 116–128) is located at the conserved CD domain, and the other two are located at the hinge region: a bipartite NLS2 (GKHRK-LERS, amino acids 165–188) and NLS3 (RCRGS-VKRF, amino acids 324–339). Then, we constructed the GmLHP1 deletion mutants (GmLHP1-1 to 8), each fused with GFP at its C terminus, and analyzed its subcellular localization (Fig. 6a). Transient expression into Arabidopsis protoplasts showed

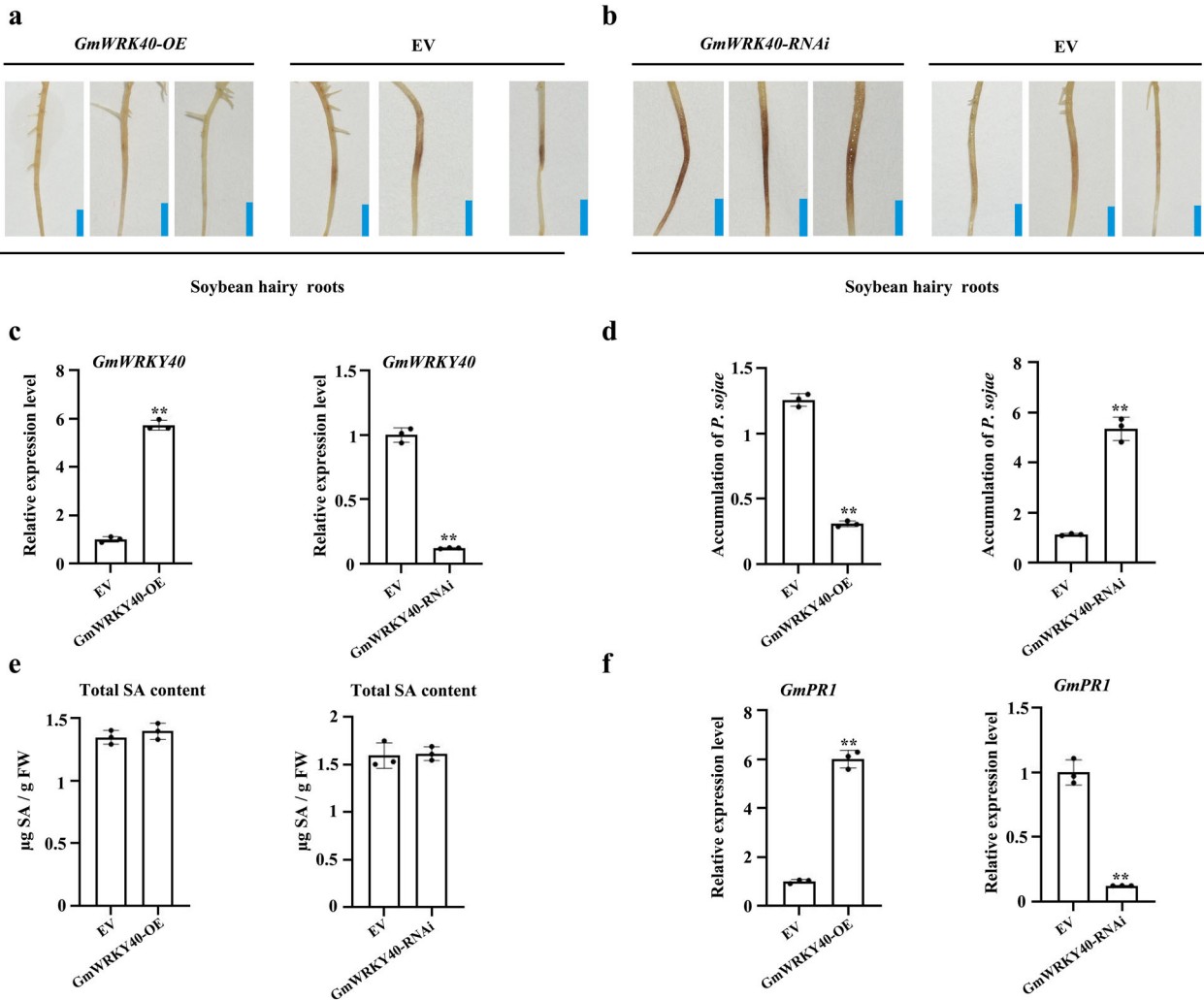

**Fig. 5 GmWRKY40 also functions downstream of SA biosynthesis and enhances the expression of SA-marker gene *GmPR1* in response to *P. sojae*.**
**a** Typical phenotypes of *WRKY40-OE* and EV soybean hairy roots after 48 h of *P. sojae* inoculation. Bars, 0.5 cm. **b** Typical phenotypes of *WRKY40-RNAi* and EV soybean hairy roots after 48 h of *P. sojae* inoculation. Bars, 0.5 cm. **c** qRT-PCR analysis of relative *GmLHP1* expression in transgenic soybean hairy roots. Soybean hairy roots transformed with empty vector (EV) were used as controls; the expression level of the control sample (EV) was set to 1. **d** Relative biomass of *P. sojae* in *GmLHP1*-transgenic soybean hairy roots based on the transcript level of *P. sojae TEF1* (EU079791). **e** SA contents in *WRKY40-OE*, *WRKY40-RNAi*, and EV hairy roots. FW, fresh weight. **f** Relative expression level of *GmPR1* in *WRKY40-OE*, *WRKY40-RNAi*, and EV hairy roots. The expression level of the control sample (EV) was set to 1. The housekeeping gene *GmEF1* was used as an internal control to normalize the data. The experiment was performed on three biological replicates, each with three technical replicates, and the results were statistically analyzed using Student's *t*-test (*$P < 0.05$, **$P < 0.01$). Bars indicate the standard deviation of the mean ($n = 3$).

that the localization of GmLHP1-1 (amino acids 108–448), containing all NLS regions but lacking the N-terminal part (amino acids 1–107), and GmLHP1-2 (amino acids 1–373), containing all NLS regions but lacking the CSD domain, both were localized in the nucleus of the transformed cell and were indistinguishable from that of the intact protein (amino acids 1–448). The results showed that the absence of N-terminal part alone or the CSD domain alone does not change the nuclear localization of GmLHP1. GmLHP1-3 (amino acids 340–448), containing the conserved CSD domain region, green fluorescent signal was dispersed in the entire cell of protoplasts similar to that displayed by GFP alone, further indicating that the conserved CSD domain has no specific nuclear targeting properties. GmLHP1-4 (amino acids 1–323), containing the conserved CD domain and NLS2 regions but lacking NLS3 and the CSD domain, was localized in the nucleus of the transformed cell, while GmLHP1-5 (amino acids 189–448), which containing NLS3 and the CSD domain, green fluorescent signal was dispersed in the entire cell of protoplasts

similar to that displayed by GFP alone, suggesting that the region (amino acids 189–448) of GmLHP1 is not required for the nuclear localization of GmLHP1 and the putative NLS3 region is non-functional. On the basis of GmLHP1-5 deletion mutant sequence (amino acids 189–448), GmLHP1-6 (amino acids 165–448) which added NLS2 region was localized in the nucleus, suggesting NLS2 region has specific nuclear targeting properties. Furthermore, we found that the region encompassing residues 108 to 164 (GmLHP1-7), corresponding to NLS1 region, retained the nucleolus-targeting localization property, indicating the putative NLS1 may also be functional, like the NLS2 region. To verify this prediction, we finally constructed the GmLHP1-8 deletion mutants (amino acids 1–115~129–164~189–448), which deleted the NLS1 and NLS2, and after transformation, we analyzed its subcellular localization. GmLHP1-8 was detected in the entire cell of protoplasts similar to the GFP alone control. Together, the results showed that both NLS1 and NLS2 regions are required for the nuclear targeting properties of GmLHP1.

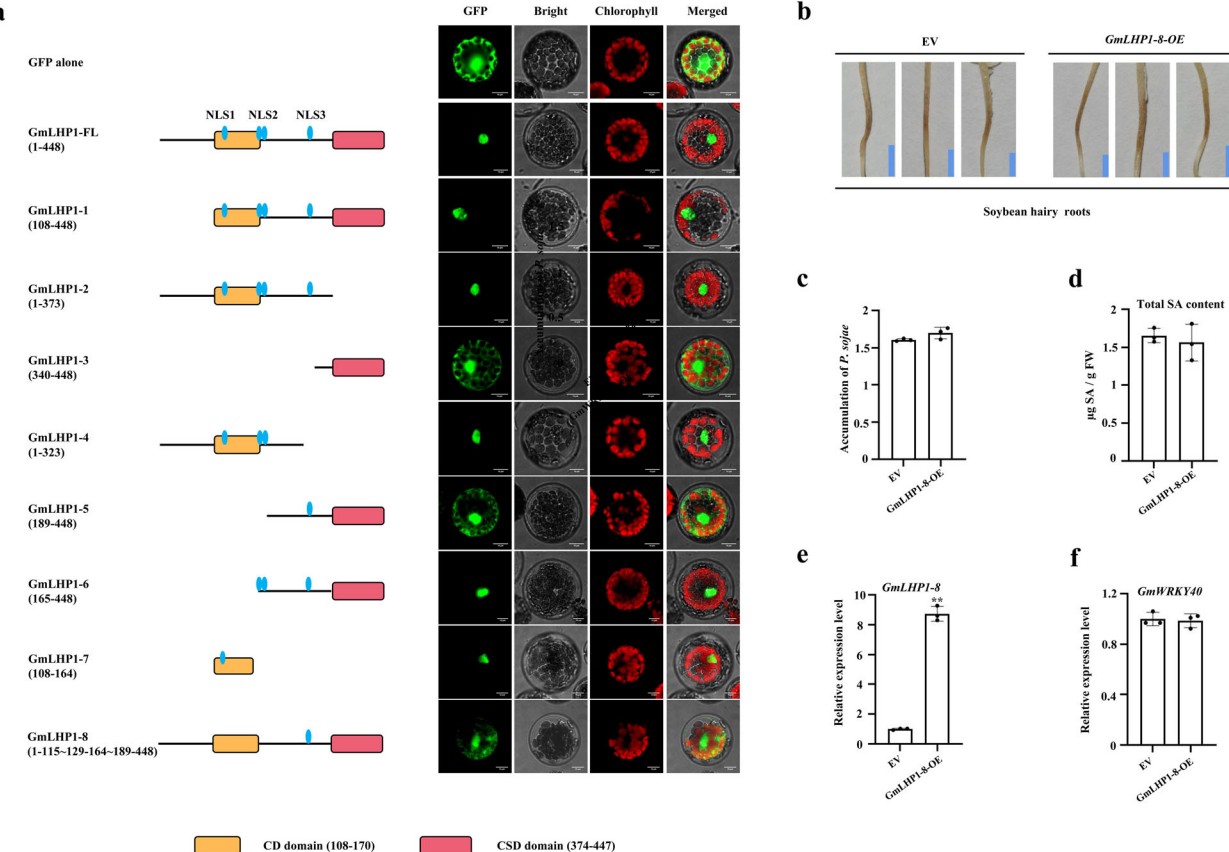

**Fig. 6 Deletion analysis of GmLHP1. a** Subcellular localization of various GmLHP1 deletion mutants. Left column, scheme of GmLHP1 and its deletion mutants fused to GFP. FL, full length. NLS, nuclear localization signal. Right column, the indicated constructs were transiently expressed in Arabidopsis protoplasts and inspected with a confocal microscope. Scale bars indicate 10 μm. **b** Typical infection symptoms of *GmLHP1-8-OE* and EV soybean hairy roots at 48 h after *P. sojae* inoculation. Bars, 0.5 cm. **c** Relative biomass of *P. sojae* in *GmLHP1-8-OE* hairy roots based on *P. sojae TEF1* (EU079791) transcript levels. **d** SA contents in *GmLHP1-8-OE* hairy roots. FW, fresh weight. **e** *GmLHP1-8* transcript levels in EV and *GmLHP1-8-OE* hairy roots. **f** GmWRKY40 transcript levels in EV and *GmLHP1-8-OE* hairy roots. The expression level of the control sample (EV) was set to 1. The housekeeping gene *GmEF1* was used as an internal control to normalize the data. The experiment was performed on three biological replicates, each with three technical replicates, and the results were statistically analyzed using Student's *t*-test (*$P < 0.05$, **$P < 0.01$). Bars indicate the standard deviation of the mean ($n = 3$).

To analyze whether the nuclear localization of GmLHP1 is necessary for its functionality, we investigated the *P. sojae* resistance in *GmLHP1-8-OE* transgenic soybean hairy roots. After 2 days of incubation with *P. sojae* zoospores, there was no significant phenotype difference between EV and *GmLHP1-8-OE* soybean hairy roots (Fig. 6b). In accordance with this, the relative biomass of *P. sojae* in infected EV and *GmLHP1-8-OE* soybean hairy roots have no significant difference (Fig. 6c). Furthermore, the SA levels in *GmLHP1-8-OE* soybean hairy roots were not significantly downregulated compared to that in EV soybean hairy roots (Fig. 6d). To further determine whether the changes of GmLHP1 nuclear localization have an effect on the suppression of *GmWRKY40* expression by GmLHP1, we also analyzed the expression levels of *GmLHP1-8* and *GmWRKY40* in *GmLHP1-8-OE* soybean hairy roots (Fig. 6e, f). *GmWRKY40* expression was not significantly suppressed in the *GmLHP1-8-OE* soybean hairy roots. These results indicated that the nuclear localization of GmLHP1 is required for its functionality.

**GmBTB/POZ releases GmLHP1-regulated *GmWRKY40* suppression in *GmLHP1-OE* soybean lines.** We also analyzed the expression levels of *GmWRKY40* in *GmBTB/POZ-OE* and *GmBTB/POZ-RNAi* soybean plants. As shown in Fig. 7a, *GmWRKY40* was upregulated in *GmBTB/POZ-OE* plants and downregulated in *GmBTB/POZ-RNAi* plants compared to the WT. These results

indicate that GmBTB/POZ is also involved in regulating *GmWRKY40* transcription. To further explore the role of GmBTB/POZ in GmLHP1-mediated suppression of *GmWRKY40* expression, we generated *GmLHP1-OE* and *GmBTB/POZ-OE/GmLHP1-OE* transgenic soybean hairy roots and used hairy roots transformed with EV as a negative control. After measuring *GmBTB/POZ* and *GmLHP1* transcript levels to evaluate the efficiency of *GmBTB/POZ* and *GmLHP1* expression (Fig. 7b, c), we measured *GmWRKY40* transcript levels in EV, *GmLHP1-OE*, and *GmBTB/POZ-OE/GmLHP1-OE* hairy roots by qRT-PCR. *GmWRKY40* expression was significantly suppressed in the *GmLHP1-OE* lines, but this effect was inhibited in *GmBTB/POZ-OE/GmLHP1-OE* hairy roots (Fig. 7d). These results suggest that GmBTB/POZ releases GmLHP1-mediated suppression of *GmWRKY40* expression, likely by inducing the degradation of GmLHP1.

**GmBTB/POZ increases resistance to *P. sojae* in *GmLHP1-OE* soybean lines.** Since GmBTB/POZ directly interacts with GmLHP1 to induce its degradation and weakens GmLHP1-mediated *GmWRKY40* suppression, we investigated whether GmBTB/POZ modifies GmLHP1-regulated *P. sojae* defense responses by quantifying *P. sojae* biomass in EV control, *GmLHP1-OE*, and *GmBTB/POZ-OE/GmLHP1-OE* transgenic soybean hairy roots at 48 hpi. As expected, *P. sojae* biomass was significantly (**$P < 0.01$) higher in *GmLHP1-OE* hairy roots than

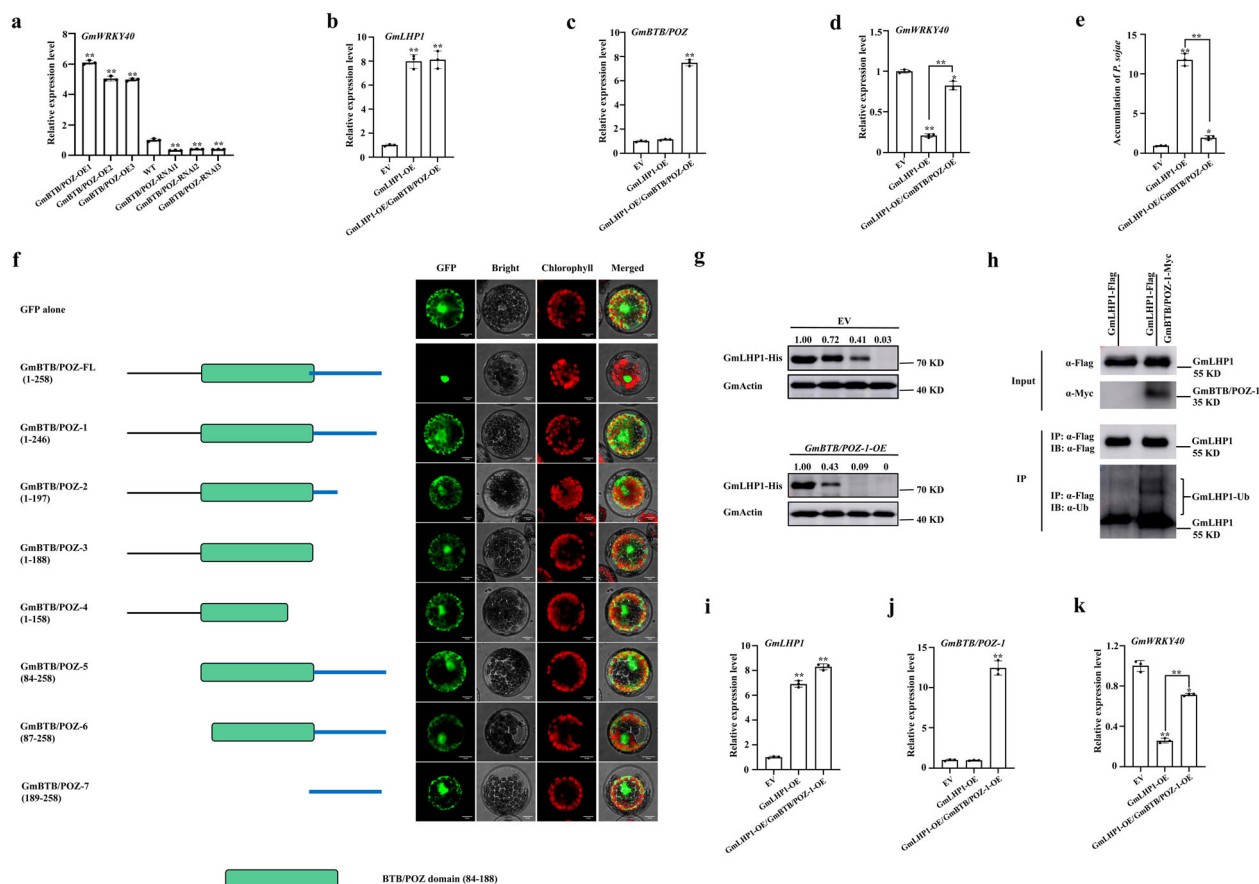

**Fig. 7 GmBTB/POZ weakens GmLHP1-mediated suppression of *GmWRKY40* and increases GmLHP1-regulated responses to *P. sojae*. a** *GmWRKY40* transcript levels in *GmBTB/POZ-OE* and *GmBTB/POZ-RNAi* soybean plants. **b** *GmLHP1* transcript levels in EV, *GmLHP1-OE*, and *GmBTB/POZ-OE/GmLHP1-OE* transgenic soybean hairy roots. **c** *GmBTB/POZ* transcript levels in EV, *GmLHP1-OE*, and *GmBTB/POZ-OE/GmLHP1-OE* transgenic hairy roots. **d** *GmWRKY40* transcript levels in EV, *GmLHP1-OE*, and *GmBTB/POZ-OE/GmLHP1-OE* transgenic hairy roots. **e** Relative biomass of *P. sojae* in EV, *GmLHP1-OE*, and *GmBTB/POZ-OE/GmLHP1-OE* transgenic hairy roots based on the transcript level of *P. sojae TEF1* (EU079791) after 48 h of *P. sojae* inoculation. **f** Subcellular localization of various GmBTB/POZ deletion mutants. Left column, scheme of GmBTB/POZ and its deletion mutants fused to GFP. FL, full length. Right column, the indicated constructs were transiently expressed in Arabidopsis protoplasts and inspected with a confocal microscope. Scale bars indicate 10 μm. **g** In vitro cell-free degradation assays of GmLHP1-His in protein extracts from *GmBTB/POZ-1-OE* soybean hairy roots. **h** GmBTB/POZ-1 promotes the ubiquitination of GmLHP1 in vivo. GmLHP1-Flag was immunoprecipitated using anti-Flag-Tag Mouse mAb (Agarose Conjugated) from *GmLHP1-OE* and *GmLHP1-OE/GmBTB/POZ-1-OE* soybean hairy roots by high-efficiency *A. rhizogenes*-mediated transformation. The transgenic hairy roots were treated with 100 μM MG132 for 8 h before extraction. The immunoprecipitated protein was examined using anti-Flag and anti-ubi antibodies. **i** *GmLHP1* transcript levels in EV, *GmLHP1-OE*, and *GmLHP1-OE/GmBTB/POZ-1-OE* transgenic soybean hairy roots. **j** *GmBTB/POZ* transcript levels in EV, *GmLHP1-OE*, and *GmLHP1-OE/GmBTB/POZ-1-OE* transgenic hairy roots. **k** *GmWRKY40* transcript levels in EV, *GmLHP1-OE*, and *GmLHP1-OE/GmBTB/POZ-1-OE* transgenic hairy roots. The housekeeping gene *GmEF1* was used as an internal control to normalize the data. The experiment was performed on three biological replicates, each with three technical replicates, and the results were statistically analyzed using Student's *t*-test (\**P* < 0.05, \*\**P* < 0.01). Bars indicate the standard deviation of the mean (*n* = 3).

in EV hairy roots (Fig. 7e). However, the overexpression of *GmBTB/POZ* resulted in a significant reduction in *P. sojae* biomass (Fig. 7e). These results indicate that GmBTB/POZ modulates GmLHP1-mediated *P. sojae* defense responses in soybean, possibly by regulating the expression of the downstream target gene *GmWRKY40*.

**Regulatory mechanism of GmBTB/POZ to GmLHP1 is independent of exclusive or predominant nuclear localization of GmBTB/POZ.** To test whether the nuclear localization of GmBTB/POZ is required for the regulatory mechanism of GmBTB/POZ to GmLHP1, we firstly analyzed the NLS regions of GmBTB/POZ using NLS Mapper software[56,57]. However, no putative NLS region was identified. We further constructed the GmBTB/POZ deletion mutants, each fused with GFP at its C terminus, and analyzed its

subcellular localization (Fig. 7f). Transient expression into Arabidopsis protoplasts showed that the full-length GmBTB/POZ protein (amino acids 1–258) was localized to the nucleus, which has also been demonstrated by Zhang et al.[35], while all the GmBTB/POZ deletion mutants (GmBTB/POZ-1 to 7) green fluorescent signal was dispersed in the entire cell of protoplasts similar to that displayed by GFP alone. These results suggested that the integrity of GmBTB/POZ may be required for the nuclear-targeting localization of GmBTB/POZ, the nuclear localization of GmBTB/POZ may not be controlled by a specific region.

Then, we take the deletion mutant GmBTB/POZ-1, in which the nuclear localization has been changed and the protein sequence is the nearest to the full-length GmBTB/POZ protein, to analyze whether the nuclear localization of GmBTB/POZ is required for the ubiquitination-regulatory of GmBTB/POZ to GmLHP1 by in vitro cell-free degradation assay and in vivo

ubiquitination assay. The results suggested that GmBTB/POZ-1 could promote the ubiquitination of GmLHP1 in vitro and in vivo (Fig. 7g, h). To further explore whether the change of GmBTB/POZ nuclear localization has an effect on the GmLHP1-mediated suppression of GmWRKY40 expression, we also measured GmWRKY40 transcript levels in EV, GmLHP1-OE, and GmLHP1-OE/GmBTB/POZ-1-OE soybean hairy roots (Fig. 7k), while GmBTB/POZ-1 and GmLHP1 transcript levels were tested to evaluate the efficiency of GmBTB/POZ-1 and GmLHP1 expression (Fig. 7i, j). GmWRKY40 expression was significantly suppressed in the GmLHP1-OE lines, but the effect was inhibited in GmLHP1-OE/GmBTB/POZ-1-OE soybean hairy roots (Fig. 7k), suggesting GmBTB/POZ-1 still can release GmLHP1-mediated suppression of GmWRKY40 expression. Taken together, these results indicated that the ubiquitination-regulatory of GmBTB/POZ to GmLHP1 may be independent of exclusive or predominant nuclear localization of GmBTB/POZ.

We further investigated the expression kinetics of GmBTB/POZ, GmLHP1, GmWRKY40, and GmPR1 in response to P. sojae. As shown in Supplementary Fig. 6, GmBTB/POZ was rapidly induced by P. sojae infection, with transcript levels peaking at 24 h. By contrast, GmLHP1 was downregulated after P. sojae infection and reached a peak within 24 h. GmBTB/POZ and GmLHP1 exhibited the opposite expression patterns in response to P. sojae. GmWRKY40 transcription was not significantly altered during the first 9 h of infection but reached a peak at 48 h. GmPR1 showed the slowest response to P. sojae infection, reaching a peak at 72 h. These findings support the notion that GmBTB/POZ and GmLHP1 play key roles in the response of soybean to P. sojae at both the transcriptional and post-translational levels.

## Discussion

Many soybean genes respond to P. sojae infection[35,58–62]. The characterization of such genes has helped elucidate the genetic mechanisms underlying defense against P. sojae infection[61–63]. However, knowledge about the regulator components in plant–pathogen interaction model and plant immunity has remained fragmented. In the present study, we demonstrated that GmLHP1 is an important component of the GmBTB/POZ-mediated SA and immune signaling pathway, providing evidence that the linkage between GmBTB/POZ and GmLHP1 is involved in the response of soybean to P. sojae attack.

Protein ubiquitination is a key mechanism that regulates immune responses[64]. BTB/POZ functions as a Ub ligase by forming a complex with CRL3 (ref. [17]). We previously demonstrated that GmBTB/POZ positively regulates disease resistance in plants, which primarily depends on SA signaling[35]. However, the components involved in GmBTB/POZ-mediated SA and defense response signaling had been unknown. In the current study, we demonstrated that GmBTB/POZ interacts with GmLHP1 in vitro and in vivo. In vitro protein degradation and in vivo ubiquitination assays suggested that GmBTB/POZ contributes to the Ub-mediated degradation of GmLHP1 through the 26S proteasome system (Fig. 2).

In addition to the roles of LHP1 in regulating flowering time and root development[33,34,42,65,66], its potential roles in plant responses to abiotic and biotic stress have been receiving increasing attention. LHP1 interacts with different proteins in different cell types to perform distinct functions[42,43]. In soybean, LHP1 interacts with GmPHD6 to regulate the expression of genes involved in salt tolerance[36]. Along with the observation that GmBTB/POZ interacts with and ubiquitinates GmLHP1, these findings prompted us to investigate whether GmLHP1 is also involved in the response of soybean to P. sojae infection. In agreement with our speculation,

overexpression and RNA interference analysis of transgenic soybean plants and hairy roots revealed that GmLHP1 negatively regulates the defense responses of soybean to P. sojae infection (Fig. 3). We also analyzed the SA content and expression levels of SA-marker gene GmPR1 in GmLHP1OE, WT, and GmLHP1RNAi soybean plants. Compared to WT plants, SA content and GmPR1 transcript levels were significantly lower in GmLHP1-OE plants but higher in GmLHP1-RNAi plants (Fig. 3c, d). Similar results were obtained for GmLHP1-OE and GmLHP1-RNAi transgenic soybean hairy roots (Fig. 3h, i).

SA mediates the plant immune response SAR, a long-lasting, broad-spectrum resistance response to a variety of pathogenic fungi, bacteria, and viruses[48,50,67]. SAR is characterized by increased endogenous SA levels and the increased expression of PR genes, such as PR1, which are considered to be effector genes for SAR[48]. Germinating soybean in red light improves resistance to Pseudomonas putida 229 by regulating SA levels and upregulating PR1 (ref. [68]). Consistent with this, our findings suggest that GmLHP1 negatively regulates the response of soybean to P. sojae, possibly by suppressing SA levels and GmPR1 gene expression. Our study provides clear evidence for the linkage between a BTB/POZ-mediated ubiquitination pathway and a plant LHP1-associated defense system. Such a linkage has not been previously reported for any plant species.

LHP1 represses the transcription of numerous genes, including FLOWERING LOCUS C (FLC) and the floral organ identity genes AGAMOUS (AG) and APETALA3 (AP3)[29,69,70]. It has also been reported that GmPHD6 could form a complex with LHP1 to bind to the GAL4 element through BD-GmPHD6 and to activate gene expression in soybean, indicating that LHP1 could also function as the coactivator in transcriptional complex[36]. However, in the current study, a series of physiological and biochemical assays showed that GmLHP1 could directly target and suppress the expression of GmWRKY40. In a transient expression assay in yeast cells using a GAL4-responsive reporter system, GmLHP1 alone did not activate the transcription of the reporter gene (Supplementary Fig. 4b). RNA-Seq showed that various stress-related genes, including GmWRKY40, were significantly down-regulated in GmLHP1OE transgenic soybean plants (Fig. 4a). Furthermore, qRT-PCR analysis indicated that the changes in GmWRKY40 expression were much more pronounced in GmLHP1OE and GmLHP1RNAi vs. WT plants: GmWRKY40 expression was dramatically reduced in GmLHP1OE vs. WT plants, and in GmLHP1RNAi soybean plants, GmWRKY40 expression significantly increased (**$P < 0.01$) compared to the WT, while none of the other genes showed markedly altered expression (Fig. 4b). A dual effector–reporter system using GmLHP1 as the effector and the luciferase gene under the control of the GmWRKY40 promoter as the reporter, as well as ChIP-qPCR assays, demonstrated that GmLHP1 directly binds to the GmWRKY40 promoter and suppress its expression (Fig. 4c–f).

WRKY family genes are involved in SA signaling pathways. Several WRKY genes are associated with SA biosynthesis; for example, the Arabidopsis wrky54 wrky70 double mutant has strongly increased SA levels[71]. In addition, several WRKYs are induced by SA and function downstream of SA the biosynthesis pathway. SA induces the rapid expression of WRKY genes in a number of plants[72–74]. In Arabidopsis, 49 of the 72 WRKY genes examined were differentially regulated in plants after treatment with SA[72]. In the current study, we determined that GmWRKY40 contains the WRKY domain, a highly conserved structural domain (Supplementary Fig. 7). GmWRKY40 expression was significantly induced by SA, and the amount of histochemical GUS staining in soybean hairy roots under SA treatment was clearly higher relative to mock ($H_2O$) conditions (Fig. 4g, h). Moreover, whereas SA levels in GmWRKY40 transgenic hairy

roots were not significantly different from those of the control (Fig. 5e), *GmWRKY40* expression enhanced the expression of SA-marker gene *GmPR1* (Fig. 5f). These results suggest that *GmWRKY40* functions as a SA-induced gene in the SA signaling pathway downstream of SA biosynthesis. We also demonstrated that GmLHP1 participates in the SA signaling pathway and inhibits SA accumulation. Meanwhile, exogenous SA application weakened the inhibition of *GmWRKY40* expression in *GmLHP1OE* soybean plants, suggesting that GmLHP1-mediated suppression of *GmWRKY40* expression might also occur via impaired SA accumulation (Fig. 4i, g). These findings indicate that *GmWRKY40* is a GmLHP1 target and that at least two types of mechanisms (directly repressed *GmWRKY40* expression and impaired SA accumulation) contribute to the regulation of *GmWRKY40* expression by GmLHP1.

Specific WRKY transcription factors function in plant defense responses by affecting the expression of *PR1* (ref. [75]). For instance, Arabidopsis *WRKY18* and *WRKY70* activate the expression of genes including *PR1* and increase resistance to pathogens[76,77]. Consistent with this finding, in the current study, *GmWRKY40* expression enhanced resistance to *P. sojae* (Fig. 5a–d) and increased the transcript level of *GmPR1* (Fig. 5f). Thus, perhaps GmLHP1 represses the expression of *GmWRKY40*, thereby negatively regulating resistance to *P. sojae*. Thereinto, the nuclear localization of GmLHP1 is required for the GmLHP1-mediated negative regulation of immunity, SA levels, and the suppression of *GmWRKY40* expression (Fig. 6a–f).

More importantly, *GmWRKY40* was upregulated in *GmBTB/POZ-OE* soybean lines and downregulated in *GmBTB/POZ-RNAi* lines compared to WT plants, indicating that GmBTB/POZ also affects the transcription of *GmWRKY40*. Analysis of soybean hairy roots co-transformed with GmBTB/POZ and GmLHP1 indicated that GmBTB/POZ released GmLHP1-regulated *GmWRKY40* suppression and increased resistance to *P. sojae* in *GmLHP1-OE* hairy roots.

Finally, we demonstrated that GmBTB/POZ and GmLHP1 are both involved in regulating *P. sojae* resistance and *GmWRKY40* expression but play opposite roles in this process. Specifically, we propose that GmBTB/POZ and GmLHP1 function together in SA and immune signaling pathways and that GmBTB/POZ recruits and degrades GmLHP1, thereby regulating the expression of downstream target gene *GmWRKY40* in soybean. The expressions of *GmLHP1* and *GmBTB/POZ* are inversely regulated during *P. sojae* infection (Supplementary Fig. 6). These findings strongly suggest that GmBTB/POZ and GmLHP1 play key roles in the response of soybean to *P. sojae* at both the transcriptional and post-translational levels.

Taken together, based on previous and current findings, we propose a model explaining how the GmBTB/POZ–GmLHP1 complex regulates the response of soybean to *P. sojae* infection (Fig. 8). According to our model, GmBTB/POZ and GmWRKY40 act as positive regulators, but GmLHP1 acts as a negative regulator, of the response of soybean to *P. sojae* infection. GmLHP1 functions as an upstream regulator to repress *GmWRKY40* expression by directly suppressing its promoter activity and impairing SA accumulation, thus inhibiting plant defense responses. Moreover, *P. sojae* induces the transcription of *GmBTB/POZ*, whereas *GmLHP1* is downregulated during *P. sojae* infection. The high levels of GmBTB/POZ recruit and degrade GmLHP1, thereby releasing its suppressive effect on *GmWRKY40* expression, thus increasing the defense response to *P. sojae*. This study provides compelling evidence for the role of the GmBTB/POZ–GmLHP1 complex in modulating the response of soybean to *P. sojae* infection. Furthermore, it has been previously proved that LHP1 plays a central role in regulating flowering time, and Arabidopsis loss-of-function *lhp1* mutants exhibit photoperiod-

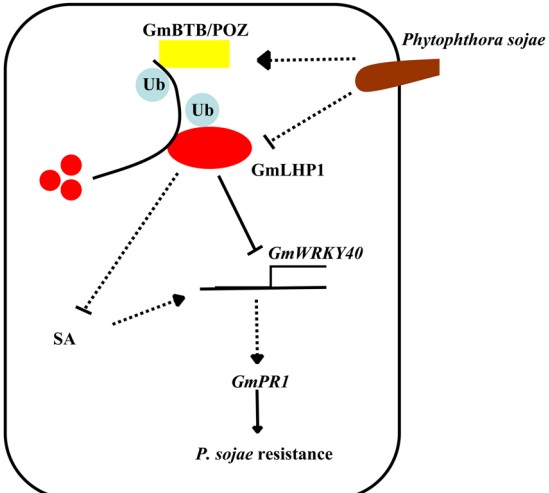

**Fig. 8 Model of the role of *GmBTB/POZ*, *GmLHP1*, and *GmWRKY40* expression in the response of soybean to *Phytophthora sojae* infection.** *P. sojae* induces the transcription of *GmBTB/POZ*, whereas *GmLHP1* is downregulated during *P. sojae* infection. GmLHP1 functions as an upstream regulator to repress *GmWRKY40* expression by directly suppressing its promoter activity and impairing SA accumulation, thus inhibiting plant defense responses. The increased levels of GmBTB/POZ recruit and degrade GmLHP1, thereby releasing the GmLHP1-suppressed expression of *GmWRKY40* and thus increasing the defense response to *P. sojae*.

independent early flowering compared to WT plants[23,78]. In our study, we also observed that *GmLHP1RNAi* soybean plants showed early flowering compared with WT plants under artificial long-day conditions (Supplementary Fig. 8). However, whether GmBTB/POZ–GmLHP1 complex is also involved in flowering-regulatory, as well as the underlying genetic and molecular mechanisms still require further exploration.

## Methods

**Plant materials and pathogen inoculation**. "Dongnong 50", a soybean (*Glycine max*) cultivar susceptible to *P. sojae* race 1, was obtained from the Key Laboratory of Soybean Biology in the Chinese Ministry of Education, Harbin, and used for gene transformation experiments and expression analysis. "Suinong 10", a soybean cultivar with gene-for-gene resistance against *P. sojae* race 1, the predominant race in Heilongjiang, China[79], was used for the gene isolation and gene expression kinetics experiments. The coding sequences (CDS) of *GmLHP1* (Glyma.16G079900) and *GmWRKY40* (Glyma.15G003300) were amplified by PCR using cDNA derived from leaves of "Suinong 10" soybean as the template. The seeds were grown in a growth chamber at 25 °C and 70% relative humidity under a 16 h light/8 h dark cycle. *N. benthamiana* plants for the LCI assays and dual-luciferase assays were grown at 22 °C under a 16 h light/8 h dark photoperiod with a light intensity of $120 \, \mu E \, m^{-2} \, s^{-1}$.

*Phytophthora sojae* race 1 (PSR01) was isolated from infected soybean plants in Heilongjiang, China[79], and cultivated at 25 °C for 7 days on V8 juice agar in a polystyrene dish.

**In vitro pull-down assay**. To produce the GmLHP1-His fusion protein, the CDS of *GmLHP1* was cloned into the pET29b (+) expression vector. The recombinant fusion plasmids were transformed into *E. coli* Rosetta (DE3) cells. The fusion proteins was purified at 4 °C and quantified according to the pET System Manual. To produce the GmBTB/POZ-GST protein, the CDS of *GmBTB/POZ* was inserted into the pGEX-4T-1 expression vector and expressed in Rosetta (DE3) cells. The target protein was purified with GST resin (GE Healthcare; 17-0756-01). Pull-down was performed as described by Yang et al.[80]. The pulled-down proteins were eluted by boiling, separated by 12% SDS-PAGE, and detected by immunoblotting using anti-GST (Abmart, code number M20007S) and anti-His antibodies (Abmart, code number M20001S), respectively.

**Firefly LCI assay**. The CDS of *GmLHP1* and *GmBTB/POZ* were fused with the N-terminal and C-terminal parts of the luciferase reporter gene, respectively. *Agrobacteria* harboring the pCAMBIA1300-*GmLHP1*nLUC and pCAMBIA1300-*GmBTB/POZ*cLUC constructs were co-infiltrated into *N. benthamiana* leaves,

which were subsequently sprayed with luciferin (1 mM luciferin and 0.01% Triton X-100) and photographed using Chemiluminescence imaging (Tanon 5200) at 72 h after infiltration.

**BiFC assays and subcellular localization analysis**. For interaction studies, the gene sequences were cloned into serial pSAT6 vectors encoding N- or C-terminal-enhanced yellow fluorescent protein fragments. To determine the subcellular localization of target proteins, the target gene sequences were ligated into the pCAMBIA1302 vector under the control of the 35S promoter, generating the recombinant plasmid. The resulting constructs were used for transient assays via PEG transfection of Arabidopsis protoplasts as described by Yoo et al.[81]. Transfected cells were imaged using a TCS SP2 confocal spectral microscope imaging system (Leica, Solms, Germany).

**In vitro cell-free degradation assays**. Total proteins were extracted from WT and transgenic soybean lines with degradation buffer[82]. Each reaction contained 500 µg of soybean total proteins and 100 ng of GmLHP1-His proteins purified from *E. coli* Rosetta (DE3) cells. For the proteasome inhibitor experiments, 100 µM MG132 was added to the total proteins 60 min prior to the cell-free degradation experiment. The reactions were incubated at 22 °C. The mixed solutions were collected at the designated time point (0, 0.5, 1, and 3 h) and examined using an anti-His antibody (Abmart, code number M20001S). The quantified results were analyzed using ImageJ software (https://imagej.nih.gov/ij/index.html).

**In vivo ubiquitination assay**. To detect ubiquitination of GmLHP1 in vivo, a plant binary expression vector system was constructed and used to generate *GmBTB/POZ*-OE, *GmLHP1*-OE, *GmLHP1*-OE/*GmBTB/POZ*-OE, *GmLHP1*-OE/*(domain+C)-OE*, *GmLHP1*-OE/*(N+domain)-OE*, or *GmLHP1*-OE/*GmBTB/POZ-1*-OE transgenic soybean hairy roots by high-efficiency *A. rhizogenes*-mediated transformation. The transgenic hairy roots were treated with 100 µM MG132 for 8 h prior to protein extraction. GmLHP1-Flag protein was immunoprecipitated using anti-Flag-Tag Mouse mAb (Agarose Conjugated) (Abmart, code number M20018S). The eluted proteins were detected using anti-Flag antibody (Abmart, code number M20008M) and anti-Ubi antibody (Abcam, code number ab19169).

**Plasmid construction and genetic transformation of soybean**. To produce the GmLHP1 overexpression and GmLHP1-Flag fusion constructs, the CDS of *GmLHP1* and Flag sequence (ATGGACTACAAGGATGACGATGACAAG) were cloned into the pCAMBIA3301 vector with the *bar* gene (as the selectable marker) and Flag tag under the control of the cauliflower mosaic virus 35S (CaMV35S) promoter to overexpress *GmLHP1*. To suppress *GmLHP1* expression, the cDNA fragment of *GmLHP1* was amplified and inserted into vector pFGC5941 (ref. [83]). The *p35S: Flag-GmLHP1* and *p35S: GmLHP1-RNAi* recombinant plasmids were transferred into *Agrobacterium tumefaciens* strain LBA4404 via the freeze–thaw method as described by Holsters et al.[84]. "Dongnong 50" soybean was used for the gene transformation experiments, and transgenic soybean plants expressing *p35S: Flag-GmLHP1* and *p35S: GmLHP1-RNAi* were generated by *Agrobacterium*-mediated transformation using the cotyledonary node method[85]. All primers used for genotyping and vector construction are listed in Supplementary Table 1.

**Agrobacterium rhizogenes-mediated transformation of soybean hairy roots**. To construct the *p35S: Flag-GmWRKY40* overexpression vector, the CDS of *GmWRKY40* was cloned into plant expression vector pCAMBIA3301 with Flag tag as the selectable marker. To construct the *GmWRKY40* RNAi vector, the cDNA fragment of *GmWRKY40* was amplified and inserted into vector pFGC5941 (ref. [83]). Transgenic soybean hairy roots were generated by *A. rhizogenes*-mediated transformation as described by Graham et al.[44] and Kereszt et al.[45], with some modifications.

**Assessment of soybean disease responses and SA levels**. For phenotypic analysis of the response of soybean to *P. sojae* infection, artificial inoculation was performed as described by Dou et al.[86] and Ward et al.[87] with minor modifications. Soybean roots and hairy roots were inoculated with *P. sojae* zoospores (approximately $1 \times 10^5$ spores mL$^{-1}$). Disease symptoms on each root were observed after inoculation and photographed with a Nikon B7000 camera. SA levels were determined as described by Aboul et al.[88] and Pan et al.[89].

**RNA-Seq analysis**. Three independent *GmLHP1-OE* transgenic soybean plants and three WT "Dongnong 50" plants grown for 6 weeks in the field under non-stress conditions were used for RNA-Seq analysis. Sequencing libraries were generated using a NEB Next Ultra RNA Library Prep Kit for Illumina (NEB, USA) following the manufacturer's recommendations, and index codes were added to each sample. After cluster analysis, the RNA samples were sequenced on the Illumina HiSeq 2500 platform to generate paired-end reads. Total reads were mapped to the soybean genome using TopHat software. Read counts for each gene were generated using HTSeq in union mode. DEGs between samples were defined by DESeq using two separate models[90], based on fold change >2 and FDR-adjusted

$P$ value < 0.05. GO enrichment analysis of the DEGs was performed using the GOseq R packages based on Wallenius noncentral hypergeometric distribution[91], which adjusts for gene length bias in DEGs.

**ChIP assay**. For the ChIP assays, WT and *p35S: Flag-GmLHP1* transgenic plants were subjected to chromatin extraction and immunoprecipitation as described by Saleh et al.[92]. Briefly, the leaves from 30-day-old plants were harvested for fixation. Nuclei were isolated and sonicated to generate DNA fragments with an average size of 500 bp. The soluble chromatin fragments were isolated and pre-absorbed with 30 µL of anti-Flag-Tag Mouse mAb (Agarose Conjugated) (Abmart, code number M20018S) to eliminate nonspecific binding and immunoprecipitated by 30 µL of anti-Flag-Tag Mouse mAb (Agarose Conjugated) (Abmart, code number M20018S). The precipitated DNA was recovered and analyzed by quantitative PCR with SYBR Premix ExTaq Mix (Takara, Japan). The ChIP-qPCR results are reported as relative binding units (IP/Input). The primers used are listed in Supplementary Table 1.

**Transient transcription dual-luciferase assay**. The promoter region of *GmWRKY40* was amplified, cloned into the pGreenII 0800-LUC vector, and used as a reporter[93]. *p35S: Flag-GmLHP1* was used as an effector construct. The effector and reporter constructs were cotransfected into healthy leaves of 21-day-old *N. benthamiana* plants by agroinfiltration[94]. The plants were incubated under continuous white light for 3 days after infiltration, sprayed with luciferin (1 mM luciferin and 0.01% Triton X-100), and photographed using Chemiluminescence imaging (Tanon 5200) at 72 h after infiltration. Leaf samples were collected for the dual-luciferase assay using a commercial kit (Promega; PR-E1910). Firefly luciferase (LUC) and Renilla luciferase (REN) activities were measured in the samples. The REN gene driven by the 35S promoter in the pGreenII 0800-LUC vector was used as an internal control. LUC activity was normalized to REN activity, and LUC/REN ratios were calculated. The data presented are the averages of at least three independent replicates.

**Statistics and reproducibility**. All statistical methods are annotated in the figure captions. The numbers of biological replicates in each assays are also indicated in the figure captions. The experiment was performed on three biological replicates, each with three technical replicates, and the results were statistically analyzed using Student's *t*-test. A difference was considered to be statistically significant when *$P < 0.05$ or **$P < 0.01$. Bars indicate the standard deviation of the mean.

**Reporting summary**. Further information on experimental design is available in the Nature Research Reporting Summary linked to this paper.

## Data availability
All data supporting the findings of this study are available in the main text and its Supplementary Information. All the source data for graphs in Figures and Supplementary Information are presented in Supplementary Data 1 and 2. Raw images of the western blots are provided in Supplementary Fig. 9. Raw RNA sequencing data are available at the NCBI Sequence ReadArchive (SRA) under accession PRJNA702619. Gene sequences, involved in this study, were obtained from Phytozome (https://phytozome.jgi.doe.gov/). The accession numbers of genes are as follows: *GmLHP1* (Glyma.16G079900), *GmBTB/POZ* (Glyma.04G244900), *GmMEKK2* (Glyma.17G173000), *GmWRKY40* (Glyma.15G003300), *GmCPK2* (Glyma.11G206300), *GmNAC90* (Glyma.11G182000), *GmNAC29* (Glyma.02G109800), *GmERF104* (Glyma.20G070000), *GmbHLH35* (Glyma.13G101100), *GmMYB70* (Glyma.17G237900), and *GmMLP34* (Glyma.09G102400).

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

## Acknowledgements
This work was supported by NSFC Projects (31671719, 31971972), Natural Science Foundation of Heilongjiang Province (ZD2019C001) and Outstanding Talents and Innovative Team of Agricultural Scientific Research.

## Author contributions
S.Z., P.X., and C.Z. designed the experiments. C.Z., Q.C., H.W., and H.G. performed the experiments. C.Z., X.F., X.C., M.Z., W.W., B.S., S.L., and J.W. analyzed the data. S.Z., P.X., and C.Z. wrote the manuscript. All of the authors read and approved the final manuscript.

## Competing interests
The authors declare no competing interests.
