## [Peer Review File · Communications Biology]

Reviewers' comments:

Reviewer #1 (Remarks to the Author):

This manuscript describes the positive regulation of plant immunity to *Phytophthora sojae* infection by previously identified soybean GmBTB/POZ. By using a range of experimental approaches, the author showed that GmBTB/POZ targets directly with soybean GmLHP1 and promotes its ubiquitination and degradation. GmLHP1 was shown to negatively regulate soybean immunity to *P. sojae*, by targeting GmWRKY40, a SA-induced transcription factor gene in the SA signaling pathway. GmLHP1 represses GmWRKY40 expression via at least two mechanisms, directly binding to its promoter and impairing SA biosynthesis. The authors further showed that GmBTB/POZ overexpression released GmLHP1-mediated GmWRKY40 suppression and increased resistance to *P. sojae*. These findings uncover a regulatory mechanism by which GmBTB/POZ-GmLHP1 modulates resistance to *P. sojae* in soybean, likely by regulating the expression of downstream target gene GmWRKY40.

The manuscript is well written and the experiments were generally carefully designed. The obtained results represent significant progress in understanding disease resistance. The research is potentially of interest and therefore merits consideration for publication. However, there are some problems with current version of this manuscript.

1. The introduction needs to be shortened, too much description of PTI and ETI that are not so much relevant to the research topic.
2. In Fig. 2, was it mislabeled that GmLHP1 was detected by anti-His antibody? I noticed that nearly all other experiments used Flag tag fusion with GmLHP1.
3. GmLHP1 is a member of nucleus localized conserved protein, its confirmation experiments need to be shortened and the results (Fig.4a) can be moved to the supplementary information.
4. Major evidence obtained in this research is based on over expression or silencing of GmLHP1 and GmBTB/POZ. It's useful to examine whether mutations in conserved domain sites or changes in localization abolish their functionality, particularly negative regulation of immunity, SA biosynthesis and suppression of GmWRKY40 expression by GmLHP1.
5. Does changed localization of GmBTB/POZ promote the ubiquitination and degradation of GmLHP1, release GmLHP1-regulated GmWRKY40 suppression?
6. LHP1 is a conserved protein, does GmLHP1 silencing cause any developmental defects?
7. RNA-Seq analysis was performed for both WT and GmLHP1OE transgenic soybean plants, however, why the plants were grown in the field where it's difficult to control the environment?
8. The Discussion section needs to be focused, too much repetition from introduction, information irrelevant to this research.
9. Too many references, shortening Introduction and Discussion can reduce good number of less-relevant references.
10. The authors need to check for correct use of italic names of genes and species.

Reviewer #2 (Remarks to the Author):

In this manuscript, Zhang, et al. demonstrate that GmBTB/POZ, a soybean Cul3 E3 ubiquitin ligase

substrate adapter protein, interacts with GmLHP1. They go on to show that it promotes ubiquitination of this target in vivo and degradation in vitro. The authors find that loss of GmLHP1 results in enhanced resistance to *P. sojae*, and that this is likely due to increased levels of SA and de-repression of the transcription factor GmWRKY40.

This study is well-conceived, and the results are generally robust. However, I do have some major concerns about the novelty and validation of some of the results.

1. The interaction between GmBTB/POZ and Gm LHP1 was already demonstrated (although not using as many approaches) in previous work from some of the authors (Zhang, et al., 2019 Mol Plant Pathol), yet the current manuscript claims, “ in the current study, we identified a GmBTB/POZ-interacting partner, designated GmLHP1”.
 2. The lower 2/3 of Fig.1 panel d (the controls) are identical to Fig S1 from the previous manuscript cited above.
 3. Showing the presence of resistance marker is not sufficient validation of the GmBTB/POZ-RNAi plants. Reduced transcript levels must be shown.
 4. According to reference 52 (Wei, et al., 2017), there are two genes encoding copies of GmLHP1. There is no mention of this, or of which one is over-expressed/targeted by RNAi.
 5. Reduced transcripts in the GmLHP1-RNAi plants must be shown.
 6. Fig 4a, indicates a lack of transcriptional activation activity for GmLHP1. This is opposite of what was reported in ref. 52. Can the authors please address this discrepancy?
 7. Fig S3 is illegible.
 8. Line 324 - an effect on SA biosynthesis was not demonstrated. Differential accumulation of SA does not necessarily reflect a change in biosynthesis.
 9. According to the methods, all experiments were performed at least three times, is the data presented a summary of all experiments?
- Additional concerns
10. Line 170 – BTB/POZ itself does not have ligase activity.
 11. Fig. 2k, were higher MW species (indicative of ubiquitination) observed in Western probed with anti-flag?
 12. Line 311 – Not all WRKY genes are SA-inducible
 13. Line 333 – I think the authors mean lower, not higher.
 14. Line 358 – Do the authors mean GmWRKY40 instead of GmLHP1?
 15. Line 438 – In soybean, not Arabidopsis.
 16. Lines 477 – 479 – what changes in expression are being referred to here?

Reviewers' comments:

Reviewer #1 (Remarks to the Author):

This manuscript describes the positive regulation of plant immunity to *Phytophthora sojae* infection by previously identified soybean GmBTB/POZ. By using a range of experimental approaches, the author showed that GmBTB/POZ targets directly with soybean GmLHP1 and promotes its ubiquitination and degradation. GmLHP1 was shown to negatively regulate soybean immunity to *P. sojae*, by targeting GmWRKY40, a SA-induced transcription factor gene in the SA signaling pathway. GmLHP1 represses GmWRKY40 expression via at least two mechanisms, directly binding to its promoter and impairing SA biosynthesis. The authors further showed that GmBTB/POZ overexpression released GmLHP1-mediated *GmWRKY40* suppression and increased resistance to *P. sojae*. These findings uncover a regulatory mechanism by which GmBTB/POZ-GmLHP1 modulates resistance to *P. sojae* in soybean, likely by regulating the expression of downstream target gene *GmWRKY40*.

The manuscript is well written and the experiments were generally carefully designed. The obtained results represent significant progress in understanding disease resistance. The research is potentially of interest and therefore merits consideration for publication. However, there are some problems with current version of this manuscript.

1. The introduction needs to be shortened, too much description of PTI and ETI that are not so much relevant to the research topic.
2. In Fig. 2, was it mislabeled that GmLHP1 was detected by anti-His antibody? I noticed that nearly all other experiments used Flag tag fusion with GmLHP1.
3. GmLHP1 is a member of nucleus localized conserved protein, its confirmation experiments need to be shortened and the results (Fig.4a) can be moved to the supplementary information.
4. Major evidence obtained in this research is based on over expression or silencing of GmLHP1 and GmBTB/POZ. It's useful to examine whether mutations in conserved domain sites or changes in localization abolish their functionality, particularly negative regulation of immunity, SA biosynthesis and suppression of *GmWRKY40* expression by GmLHP1.
5. Does changed localization of GmBTB/POZ promote the ubiquitination and degradation of GmLHP1, release GmLHP1-regulated *GmWRKY40* suppression?
6. LHP1 is a conserved protein, does GmLHP1 silencing cause any developmental defects?
7. RNA-Seq analysis was performed for both WT and *GmLHP1OE* transgenic soybean plants, however, why the plants were grown in the field where it's difficult to control the environment?
8. The Discussion section needs to be focused, too much repetition from introduction, information irrelevant to this research.

9. Too many references, shortening Introduction and Discussion can reduce good number of less-relevant references.

10. The authors need to check for correct use of italic names of genes and species.

Reviewer #2 (Remarks to the Author):

In this manuscript, Zhang, et al. demonstrate that GmBTB/POZ, a soybean Cul3 E3 ubiquitin ligase substrate adapter protein, interacts with GmLHP1. They go on to show that it promotes ubiquitination of this target in vivo and degradation in vitro. The authors find that loss of GmLHP1 results in enhanced resistance to *P. sojae*, and that this is likely due to increased levels of SA and de-repression of the transcription factor GmWRKY40.

This study is well-conceived, and the results are generally robust. However, I do have some major concerns about the novelty and validation of some of the results.

1. The interaction between GmBTB/POZ and Gm LHP1 was already demonstrated (although not using as many approaches) in previous work from some of the authors (Zhang, et al., 2019 Mol Plant Pathol), yet the current manuscript claims, “ in the current study, we identified a GmBTB/POZ-interacting partner, designated GmLHP1”.

2. The lower 2/3 of Fig.1 panel d (the controls) are identical to Fig S1 from the previous manuscript cited above.

3. Showing the presence of resistance marker is not sufficient validation of the GmBTB/POZ-RNAi plants. Reduced transcript levels must be shown.

4. According to reference 52 (Wei, et al., 2017), there are two genes encoding copies of GmLHP1. There is no mention of this, or of which one is over-expressed/targeted by RNAi.

5. Reduced transcripts in the GmLHP1-RNAi plants must be shown.

6. Fig 4a, indicates a lack of transcriptional activation activity for GmLHP1. This is opposite of what was reported in ref. 52. Can the authors please address this discrepancy?

7. Fig S3 is illegible.

8. Line 324 - an effect on SA biosynthesis was not demonstrated. Differential accumulation of SA does not necessarily reflect a change in biosynthesis.

9. According to the methods, all experiments were performed at least three times, is the data presented a summary of all experiments?

Additional concerns

10. Line 170 – BTB/POZ itself does not have ligase activity.

11. Fig. 2k, were higher MW species (indicative of ubiquitination) observed in Western probed with anti-flag?
12. Line 311 – Not all WRKY genes are SA-inducible
13. Line 333 – I think the authors mean lower, not higher.
14. Line 358 – Do the authors mean GmWRKY40 instead of GmLHP1?
15. Line 438 – In soybean, not Arabidopsis.
16. Lines 477 – 479 – what changes in expression are being referred to here?

We appreciate the reviewers' insightful suggestions and comments, all of which are very helpful to the improvement of our manuscript. Below, we have listed our point-by-point responses.

Reviewer #1 (Remarks to the Author):

This manuscript describes the positive regulation of plant immunity to *Phytophthora sojae* infection by previously identified soybean GmBTB/POZ. By using a range of experimental approaches, the author showed that GmBTB/POZ targets directly with soybean GmLHP1 and promotes its ubiquitination and degradation. GmLHP1 was shown to negatively regulate soybean immunity to *P. sojae*, by targeting *GmWRKY40*, a SA-induced transcription factor gene in the SA signaling pathway. GmLHP1 represses *GmWRKY40* expression via at least two mechanisms, directly binding to its promoter and impairing SA biosynthesis. The authors further showed that *GmBTB/POZ* overexpression released GmLHP1-mediated *GmWRKY40* suppression and increased resistance to *P. sojae*. These findings uncover a regulatory mechanism by which GmBTB/POZ-GmLHP1 modulates resistance to *P. sojae* in soybean, likely by regulating the expression of downstream target gene *GmWRKY40*.

The manuscript is well written and the experiments were generally carefully designed. The obtained results represent significant progress in understanding disease resistance. The research is potentially of interest and therefore merits consideration for publication. However, there are some problems with current version of this manuscript.

1. The introduction needs to be shortened, too much description of PTI and ETI that are not so much relevant to the research topic.

Response: Thanks for your meticulous reading. The introduction has been shortened and we have deleted the description of PTI and ETI in the introduction. Moreover, we have also reorganized and compressed the introduction of our manuscript.

2. In Fig. 2, was it mislabeled that GmLHP1 was detected by anti-His antibody? I noticed that nearly all other experiments used Flag tag fusion with GmLHP1.

Response: It was not incorrectly labeled in Fig. 2a-i. In the *in vitro* protein degradation assays, protein extracts from the WT or *GmBTB/POZ*-transgenic soybean were incubated with the His-tagged GmLHP1 (GmLHP1-His) proteins purified from *Escherichia coli* Rosetta (DE3) cells at 22°C. Then, we performed an immunoblot assay using anti-His antibody to measure the abundance of GmLHP1-His protein. We are very sorry for that the description of the assay is not enough clearer. We have added the method description to the manuscript in page 6, line 152- line 156. Moreover, This method has been described in detail in the methods section of the manuscript (as shown in the manuscript in page 23, line 681-line 689).

3. GmLHP1 is a member of nucleus localized conserved protein, its confirmation

experiments need to be shortened and the results (Fig. 4a) can be moved to the supplementary information.

Response: Thanks for your professional suggestion. We have moved the results about subcellular localization of GmLHP1 to the supplementary information (as shown in Fig. S4).

4. Major evidence obtained in this research is based on over expression or silencing of GmLHP1 and GmBTB/POZ. It's useful to examine whether mutations in conserved domain sites or changes in localization abolish their functionality, particularly negative regulation of immunity, SA biosynthesis and suppression of *GmWRKY40* expression by GmLHP1.

Response: To determine the region(s) responsible for the nuclear localization of GmLHP1 and analyse whether the nuclear localization of GmLHP1 is required for its functionality, we have constructed the GmLHP1 deletion mutants (GmLHP1-1 to 8), each fused with GFP at its C terminus, and analyzed its subcellular localization (as shown in Fig. 6a). The results showed that both NLS1 and NLS2 regions are required for the nuclear targeting properties of GmLHP1. To analyse whether the nuclear localization of GmLHP1 is necessary for its functionality, we have investigated the *P. sojae* resistance in *GmLHP1-8-OE* (which contain all region sequence but lacking NLS1 and NLS2, and the protein localization have been changed) transgenic soybean hairy roots. There was no significant resistant difference between EV and *GmLHP1-8-OE* soybean hairy roots (as shown in Fig. 6b, c). Furthermore, the SA level and the *GmWRKY40* expression in *GmLHP1-8-OE* soybean hairy roots have been analyzed (as shown in Fig. 6d, e, f). The SA level in *GmLHP1-8-OE* soybean hairy roots were not significantly down-regulated compared to that in EV soybean hairy roots (Fig. 6d) and the *GmWRKY40* expression was not significantly suppressed in the *GmLHP1-8-OE* soybean hairy roots. These results suggested that the nuclear localization of GmLHP1 is required for the GmLHP1-mediated negative regulation of immunity, SA levels and the suppression of *GmWRKY40* expression.

5. Does changed localization of GmBTB/POZ promote the ubiquitination and degradation of GmLHP1, release GmLHP1-regulated *GmWRKY40* suppression?

Response: To test whether the nuclear localization of GmBTB/POZ is required for the regulatory mechanism of GmBTB/POZ to GmLHP1, we have constructed the GmBTB/POZ deletion mutants, each fused with GFP at its C terminus, and analyzed its subcellular localization (Fig. 7f). The results showed that the integrity of GmBTB/POZ may be required for the nuclear-targeting localization of GmBTB/POZ, the nuclear localization of GmBTB/POZ may not be controlled by a specific region. Then, we took the deletion mutant GmBTB/POZ-1, which the nuclear localization have been changed and the protein sequence is the nearest to the full-length GmBTB/POZ protein, to analyse whether the nuclear localization of GmBTB/POZ is required for the ubiquitination-regulatory of GmBTB/POZ to GmLHP1 by *in vitro* cell-free degradation assay and *in vivo* ubiquitination assay. The results suggested that GmBTB/POZ-1 could promotes the ubiquitination of GmLHP1 *in vitro* and *in vivo*

(Fig. 7g, h). Moreover, we have measured *GmWRKY40* transcript levels in EV, *GmLHP1-OE*, and *GmLHP1-OE/GmBTB/POZ-1-OE* soybean hairy roots to explore whether the change of GmBTB/POZ nuclear localization has an effect on the GmLHP1-mediated suppression of *GmWRKY40* expression. The results suggested that GmBTB/POZ-1 still can release GmLHP1-mediated suppression of *GmWRKY40* expression (Fig. 7k). Taken together, these results indicated that the nuclear localization of GmBTB/POZ is not required for the ubiquitination-regulatory of GmBTB/POZ to GmLHP1, and the regulatory mechanism may be independent on the nuclear localization of GmBTB/POZ.

6. LHP1 is a conserved protein, does GmLHP1 silencing cause any developmental defects?

Response: Thanks for your elaborate comments. In our study, we have observed that *GmLHP1RNAi* soybean plants showed early flowering compared with WT plants under artificial long-day conditions (Fig. S8) and we have supplemented the sentences to the discussion section of our manuscript in Page 21, line 624-Page 22, line 631. However, whether GmBTB/POZ-GmLHP1 complex is also involved in flowering-regulatory, as well as the underlying genetic and molecular mechanisms still require further explore.

7. RNA-Seq analysis was performed for both WT and *GmLHP1OE* transgenic soybean plants, however, why the plants were grown in the field where it's difficult to control the environment?

Response: Thank you for your professional comments. Because of the limit of the laboratory conditions, three independent *GmLHP1-OE* (*p35S: Flag-GmLHP1*) transgenic soybean plants and three WT 'Dongnong 50' plants (as control plants) were grown in the field for RNA-Seq analysis to preliminary identify the constitutively regulated genes by GmLHP1 under natural condition.

8. The Discussion section needs to be focused, too much repetition from introduction, information irrelevant to this research.

Response: We have deleted the repetitive and irrelevant information and reorganized the discussion section of our manuscript.

9. Too many references, shortening Introduction and Discussion can reduce good number of less-relevant references.

Response: We have reorganized and compressed the introduction and discussion section of our manuscript, and the references have been reduced greatly.

10. The authors need to check for correct use of italic names of genes and species.

Response: We have checked and corrected the italic names of genes and species in the manuscript.

Reviewer #2 (Remarks to the Author):

In this manuscript, Zhang, et al. demonstrate that GmBTB/POZ, a soybean Cul3 E3 ubiquitin ligase substrate adapter protein, interacts with GmLHP1. They go on to show that it promotes ubiquitination of this target *in vivo* and degradation *in vitro*. The authors find that loss of GmLHP1 results in enhanced resistance to *P. sojae*, and that this is likely due to increased levels of SA and **de-repression** of the transcription factor GmWRKY40.

This study is well-conceived, and the results are generally robust. However, I do have some major concerns about the novelty and validation of some of the results.

1. The interaction between GmBTB/POZ and Gm LHP1 was already demonstrated (although not using as many approaches) in previous work from some of the authors (Zhang, et al., 2019 Mol Plant Pathol), yet the current manuscript claims, “in the current study, we identified a GmBTB/POZ-interacting partner, designated GmLHP1”.

Response: Thanks for your professional comments. We are very sorry for the unprecise description. We have amended the description as follows: We previously demonstrated that GmBTB/POZ positively regulates the response of soybean to *P. sojae* infection and GmBTB/POZ interacted with GmLHP1 (LIKE HETEROCHROMATIN PROTEIN1) in a bimolecular fluorescence complementation (BiFC) assay. In soybean, there are two genes encoding copies of LHP1 (LHP1-1 and LHP1-2). In the current study, we focused on LHP1-1, namely GmLHP1 (NCBI protein no. XP_003548606; Glyma.16G079900) which contains two highly conserved structural domains: a chromo domain and a chromo shadow domain (Fig. S1). (as shown in the manuscript in Page 5, line 122-line 129).

2. The lower 2/3 of Fig.1 panel d (the controls) are identical to Fig S1 from the previous manuscript cited above.

Response: Thanks very much for your reminder. In order to avoid data duplication, we have removed the BiFC interaction results of GmLHP1 with GmBTB/POZ in Fig.1d in this study.

3. Showing the presence of resistance marker is not sufficient validation of the *GmBTB/POZ-RNAi* plants. Reduced transcript levels must be shown.

Response: The transcript levels of *GmBTB/POZ* in T4 *GmBTB/POZ-OE* and *GmBTB/POZ-RNAi* transgenic soybean plants corresponding to our study have been analyzed and shown (Fig.S2b, d).

4. According to reference 52 (Wei, et al., 2017), there are two genes encoding copies of GmLHP1. There is no mention of this, or of which one is over-expressed/targeted by RNAi.

Response: Thanks for your suggestion. We have added the descriptions to the

5. Reduced transcripts in the *GmLHP1-RNAi* plants must be shown.

Response: The transcript levels of *GmLHP1* in T4 *GmLHP1OE* and *GmLHP1RNAi* transgenic soybean plants corresponding to our study have been analyzed and shown (Fig.S2f, h).

6. Fig 4a, indicates a lack of transcriptional activation activity for GmLHP1. This is opposite of what was reported in ref. 52. Can the authors please address this discrepancy?

Response:

Effector and reporter constructs used in transcriptional activation detection (referenced from Wei et al., 2017; Figure S2)

Thanks for your elaborate suggestions. In the mentioned reference, using dual-luciferase reporter (DLR) assay system, a firefly luciferase (*LUC*) was used as a reporter and a Renilla luciferase was used as an internal control. The *LUC* gene was driven by a minimal 35S promoter connected by five copies of the GAL4 binding element which could be bind by the GAL4 DNA binding domain of BD fusion protein (the effector and reporter constructs used in transcriptional activation were shown in the mentioned reference Figure S2). BD-LHP1 enhance the expression of *LUC* gene and thus have a higher relative *LUC* activity than the control (BD), showing that LHP1 has transcriptional activation ability which can function as the coactivator to activate the GAL4 binding element-*LUC* gene bound by DNA binding domain (as shown in Fig. 2E in the mentioned reference). The transcriptional regulation activity of the GmPHD6/LHP1 complex also showed that GmPHD6 could form a complex with LHP1 to bind to the GAL4 element through BD-GmPHD6 and to activate gene expression in soybean, indicating that LHP1 could function as the coactivator in transcriptional complex.

However, in our study, we found that GmLHP1 alone did not activate the transcription of the *GAL4* reporter gene in yeast cells. A series of physiological and biochemical assays (RNA-Seq, qRT-PCR analysis, the dual effector-reporter system and ChIP-qPCR assays) showed GmLHP1 alone could directly target and suppress the expression of *GmWRKY40*. We have added the related description to the discussion section of our manuscript in Page 19, line 546-line 551.

7. Fig S3 is illegible.

Response: Thanks for your suggestion. We have provided the clear picture with volcano plots and GO functional classification results (as shown in Fig.S5).

8. Line 324-an effect on SA biosynthesis was not demonstrated. Differential accumulation of SA does not necessarily reflect a change in biosynthesis.

Response: Thanks for your professional comments. We are very sorry for the unprecise description. “SA biosynthesis” has been corrected as “SA accumulation” in Page 12, line 325 in manuscript. In addition, we have also checked and corrected other issues which are similar to the mistakes.

9. According to the methods, all experiments were performed at least three times, is the data presented a summary of all experiments?

Response: Thanks for your professional comments. All experiments were repeated at least three times with similar results, and the data were based on the average of three parallel experiments. We have added the description to the methods section of our manuscript in Page 26, line 772-line 773.

Additional concerns

10. Line 170 – BTB/POZ itself does not have ligase activity.

Response: Thanks for your professional suggestion. We have amended it as follows: BTB/POZ proteins are a bridge between CUL3-RING E3 ligase and substrate proteins, and they are essential for the ubiquitin process (as shown in the manuscript in Page 6, line 147-line 148).

11. Fig. 2k, were higher MW species (indicative of ubiquitination) observed in Western probed with anti-flag?

Response: Higher MW polypeptides can not be observed in Western blot probed with anti-Flag. However, in the *in vivo* ubiquitination assays, except for purpose protein banding, other protein bands may also be observed in Western blot probed with anti-Flag under certain conditions. Because a few other proteins may be eluted and remained after immunoprecipitated GmLHP1-Flag from proteins extracted from the plants using anti-Flag antibody, the other protein bands detected by anti-Flag could also be any other proteins. Thus, the elution proteins should be further detected using anti-Ubi antibodies to indicative ubiquitination.

12. Line 311-Not all WRKY genes are SA-inducible.

Response: Thanks for your professional suggestion. We are very sorry for the unprecise description. We have amended it as follows: Some *WRKY* genes are SA-inducible transcription factor genes involved in disease resistance in a number of plant species (as shown in the manuscript in Page 11, line 312-line 313).

13. Line 333 – I think the authors mean lower, not higher.

Response: Thanks for your meticulous reading. “higher” has been corrected as “lower” in the manuscript in Page 12, line 335.

14. Line 358 – Do the authors mean GmWRKY40 instead of GmLHP1?

Response: We are very sorry for the mistake. “GmLHP1” has been corrected as “*GmWRKY40*” in the manuscript in Page 13, line 360.

15. Line 438 – In soybean, not Arabidopsis.

Response: “In Arabidopsis” has been corrected as “In soybean” in the manuscript in Page 18, line 520. In addition, we have also checked and corrected other issues which are similar to the mistakes.

16. Lines 477 – 479 – what changes in expression are being referred to here?

Response: Thanks for your elaborate suggestions. The changes in expression referred to here are that *GmWRKY40* expression was dramatically reduced in *GmLHP1OE* vs. WT plants, and in *GmLHP1RNAi* soybean plants, *GmWRKY40* expression significantly increased (** $P < 0.01$) compared to the WT, while none of the other genes showed markedly altered expression (Fig. 4b). We have supplemented the sentences to the manuscript in Page 19, line 557-line 560.

REVIEWERS' COMMENTS:

Reviewer #3 (Remarks to the Author):

In this revised manuscript, the authors have satisfactorily met all of my previous concerns. I have one concern regarding some of the new material that has been added. I think it is incorrect to claim that the "Regulatory mechanism of GmBTB/POZ to GmLHP1 is independent on the nuclear localization of GmBTB/POZ", since the mis-localized protein is still present in the nucleus. I believe it would be more correct to say it is independent of exclusive or predominant nuclear localization.

Other minor concerns:

- In figure 7 panel g is labeled with BTB/POZ-3-OE, I believe this should be BTB/POZ-1-OE, according to the text.

- I did not see any citation for the software used to identify NLS sequences.

Response to Reviewer #3

Reviewer #3 (Remarks to the Author):

In this revised manuscript, the authors have satisfactorily met all of my previous concerns. I have one concern regarding some of the new material that has been added. I think it is incorrect to claim that the "Regulatory mechanism of GmBTB/POZ to GmLHP1 is independent on the nuclear localization of GmBTB/POZ", since the mis-localized protein is still present in the nucleus. I believe it would be more correct to say it is independent of exclusive or predominant nuclear localization.

Response: Thanks for your professional suggestion and support. "Regulatory mechanism of GmBTB/POZ to GmLHP1 is independent on the nuclear localization of GmBTB/POZ" have been corrected as "Regulatory mechanism of GmBTB/POZ to GmLHP1 is independent of exclusive or predominant nuclear localization of GmBTB/POZ" in Page 16, line 455-456 in manuscript. Moreover, we have also checked and corrected the related description in Page 17, line 485-487 in manuscript as follows: Taken together, these results indicated that the ubiquitination-regulatory of GmBTB/POZ to GmLHP1 may be independent of exclusive or predominant nuclear localization of GmBTB/POZ.

Other minor concerns:

- In figure 7 panel g is labeled with BTB/POZ-3-OE, I believe this should be BTB/POZ-1-OE, according to the text.

Response: Thanks for your meticulous reading. We are very sorry for the oversight. We have corrected it in Figure 7g.

- I did not see any citation for the software used to identify NLS sequences.

Response: We have added the corresponding references (reference 56 and 57) of NLS Mapper software in the manuscript.